# Explicit Group Sparse Projection with Applications to Deep Learning and NMF

**Riyasat Ohib**                                                     *riyasat.ohib@gatech.edu*
*TReNDS Center, Georgia Institute of Technology*

**Nicolas Gillis**                                                   *nicolas.gillis@umons.ac.be*
*University of Mons*

**Niccolò Dalmasso**                                                 *niccolo.dalmasso@jpmchase.com*
*J.P. Morgan AI Research*

**Sameena Shah**                                                     *sameena.shah@jpmchase.com*
*J.P. Morgan AI Research*

**Vamsi K. Potluru**                                                 *vamsi.k.potluru@jpmchase.com*
*J.P. Morgan AI Research*

**Sergey Plis**                                                      *s.m.plis@gmail.com*
*TReNDS Center*

**Reviewed on OpenReview:** *https://openreview.net/forum?id=jIrOeWjdpc*

## Abstract

We design a new sparse projection method for a set of vectors that guarantees a desired average sparsity level measured leveraging the popular Hoyer measure (an affine function of the ratio of the $\ell_1$ and $\ell_2$ norms). Existing approaches either project each vector individually or require the use of a regularization parameter which implicitly maps to the average $\ell_0$-measure of sparsity. Instead, in our approach we set the Hoyer sparsity level for the whole set explicitly and simultaneously project a group of vectors with the Hoyer sparsity level of each vector tuned automatically. We show that the computational complexity of our projection operator is linear in the size of the problem. Additionally, we propose a generalization of this projection by replacing the $\ell_1$ norm by its weighted version. We showcase the efficacy of our approach in both supervised and unsupervised learning tasks on image datasets including CIFAR10 and ImageNet. In deep neural network pruning, the sparse models produced by our method on ResNet50 have significantly higher accuracies at corresponding sparsity values compared to existing competitors. In nonnegative matrix factorization, our approach yields competitive reconstruction errors against state-of-the-art algorithms.

## 1 Introduction

Sparsity is a crucial property in signal processing and learning representations, exemplified by breakthroughs in compressed sensing (Donoho, 2006; Candès et al., 2006), low-rank matrix approximations (d'Aspremont et al., 2007) and sparse dictionary learning (Aharon et al., 2006; Hoyer, 2004). A natural advantage of sparseness is that it diminishes the effects of random noise since it suppresses arbitrary combinations of measured signals (Donoho, 1995; Hyvärinen, 1999; Elad, 2006). The $l_0$-norm is a natural measure of sparsity but directly optimizing for it is typically NP-hard. In practice, $\ell_1$-norm is typically used as a proxy for obtaining sparse solutions. This is similar to the LASSO problem (Tibshirani, 1996) in the regression setting where one constrains the number of non-zeros in the solution which can be accomplished explicitly as a

constraint or implicitly through $\ell_1$-norm regularization. However, this necessitates the search over the regularization parameter which corresponds to the solution with the user-defined sparsity.

In this paper, we design a new sparse projection method for a set of feature vectors $\{c_i \in \mathbb{R}^{n_i}\}_{i=1}^r$ to achieve a desired average sparsity level that is measured using the sparsity measure introduced by Hoyer (2004). For $x \neq 0$ and $n > 1$, the Hoyer sparsity of $x$ is defined as follows

$$\mathrm{sp}(x) = \frac{\sqrt{n} - \frac{\|x\|_1}{\|x\|_2}}{\sqrt{n} - 1} \ \in \ [0, 1]. \tag{1}$$

We have that $\mathrm{sp}(x) = 0 \iff \|x\|_1 = \sqrt{n}\|x\|_2 \iff |x(j)| = b$ for all $j$ and for some constant $b$, while $\mathrm{sp}(x) = 1 \iff \|x\|_0 = 1$. We refer to Section 2.1 for more detail on this measure. In summary, given a set of vectors $\{c_i \in \mathbb{R}^{n_i}\}_{i=1}^r$ and a desired average sparsity level $s \in [0, 1]$, we will compute a set of vectors $\{x_i \in \mathbb{R}^{n_i}\}_{i=1}^r$ that are closest to $\{c_i \in \mathbb{R}^{n_i}\}_{i=1}^r$ for each $i$ (see Section 3 for the details), while the average Hoyer sparsity of the vectors $x_i$'s is larger than $s$, that is, $\frac{1}{r}\sum_{i=1}^r \mathrm{sp}(x_i) \geq s$.

This projection is inspired by the work of Thom et al. (2015) in which each feature vector $c_i$ is independently projected, and used for sparse dictionary learning. Although a range of approaches exist to induce sparsity, this particular measure has been shown to enjoy many attractive properties (Hurley & Rickard, 2009) while also being very amenable for optimization. The key difference with our projection from previous works by Potluru et al. (2013); Thom et al. (2015) is that the feature vectors achieve *an average target sparsity level*: some may end up dense, while others extremely sparse based on the problem. Therefore, our approach has three main advantages:

1. Only one sparsity parameter has to be chosen, namely $s \in [0, 1]$.

2. The sparsity levels of the projected vectors are automatically tuned to achieve the desired average sparsity; hence, allowing these vectors to have different sparsity levels.

3. Our projection has more degrees of freedom and consequently will generate sparse feature vectors that are closer to the original ones.

Our novel projection operator will rely on duality and Newton's method to compute the unique solution under mild assumptions. This new approach to project a set of vectors could be used in numerous applications where more than one sparse vector has to be learned, e.g., in dictionary learning (Thom et al., 2015) and sparse low-rank matrix approximations (d'Aspremont et al., 2007). In this paper, we will explore our novel projection operator in the settings of pruning deep neural networks and sparse non-negative matrix factorization (NMF).

## 1.1 Related Work

**Projection onto the $\ell_1$ ball, and onto the intersection of $\ell_1, \ell_2$ balls** $\ell_1$-ball projections have a rich history in the literature and in particular have been considered earlier by Gafni & Bertsekas (1984). Recent versions are optimal with run times linear in size of the input (Duchi et al., 2008; Condat, 2016). Projections onto the intersection of $\ell_1, \ell_2$-ball constraints was introduced by Hoyer (2004) and subsequently addressed in a series of works by Yu et al. (2012); Potluru et al. (2013); Thom et al. (2015); Liu et al. (2019), resulting in essentially the same optimal run times as shown in the $\ell_1$ settings. Note that none of the approaches above can directly handle the group sparsity problem that arises in Sparse NMF and neural network models.

**Sparse NMF** In nonnegative matrix factorization, an input data matrix $Y \in \mathbb{R}^{m \times n}$ is approximated by a low-rank matrix $\hat{Y} = XH$ where $X \in \mathbb{R}_+^{m \times r}$ and $H \in \mathbb{R}_+^{r \times n}$, and $r$ is the factorization rank. The most popular formulation of NMF uses the Frobenius norm to evaluate the quality of the solution as follows:

$$\min_{X \in \mathbb{R}^{m \times r}, H \in \mathbb{R}^{r \times n}} \|Y - XH\|_F^2$$
$$\text{such that } X \geq 0 \text{ and } H \geq 0. \tag{2}$$

Many algorithms have been proposed to tackle this problem such as by Kim & Park (2007) and Potluru et al. (2013). Most of them use an alternating strategy optimizing X for H fixed and then H for X fixed, since the corresponding subproblems are convex; see Section 5.1 for more detail.

In practice, it is particularly useful to have sparse X and/or H to take prior information into account, leading to more robust, identifiable and more interpretable decomposition (Hoyer, 2004). We will discuss in detail sparse NMF formulations in Section 5.1.

**Neural Network Pruning**  It has been known since the 1980s that a significant number of parameters can be eliminated from neural networks without appreciable loss in accuracy (Janowsky, 1989; LeCun et al., 1990; Reed, 1993). It is indeed an attractive proposition to prune such large networks for real-time applications especially, on edge devices with resource constraints. Pruning large networks could substantially reduce the computational demands of inference when used with appropriate libraries (Elsen et al., 2020) or hardwares designed to exploit sparsity (Pool et al., 2021; Cerebras, 2019). In recent times, the Lottery Ticket Hypothesis (Frankle & Carbin, 2019) was proposed that details the presence of sub-networks within a larger network, which are capable of training to full accuracy in isolation. This resulted in a renewed interest in sparse deep learning and model pruning (Renda et al., 2020; Chen et al., 2020; 2021) and more recently in the area of sparse reinforcement learning (Arnob et al., 2021; Sokar et al., 2021). There are a range of techniques in the literature to prune deep neural networks and find sub-networks at various stages of training: techniques that prune before training (Lee et al., 2019; Wang et al., 2020), during training (Zhu & Gupta, 2018; Ma et al., 2019) and after training (Han et al., 2015). The most common among these techniques is to prune the network after training using some sort of predefined criterion that captures the significance of the parameters of the network to the objective function. A range of classical works on pruning used the second derivative information of the loss function (LeCun et al., 1990; Hassibi et al., 1993). Perhaps the most intuitive of these approaches is magnitude based pruning, where following training, a subset of the parameters below some threshold is pruned and the rest of the parameters are retrained (Han et al., 2015; Guo et al., 2016), and regularization based methods (Yang et al., 2020; Ma et al., 2019; Louizos et al., 2018; Yun et al., 2019) which induces sparsity in the network during the optimization process.

However, we still lack a way to introduce sparsity in the layers of the network which is both controllable and interpretable. For example, in the case of regularizer imposed sparsity in neural networks (Yang et al., 2020), there is no way to relate the regularization weight with the actual sparsity of the result. Moreover, in practice we have to tune the regularizer weight differently for each task and architecture. Hence a form of a grid search for that parameter is unavoidable, which is costly both in terms of time and resources.

## 1.2   Contributions and Outline

Our contributions can be summarized as follows:

1. We define a novel grouped sparse projection (GSP) with a single sparsity parameter (Section 2).

   - We provide an efficient algorithm (linear in the size of the problem) to compute this projection, based on the Newton's method (Section 3).
   - We extend our approach to perform weighted grouped sparse projection using the weighted $\ell_1$ norm (Section 4 and Appendix C).

2. We evaluate GSP on sparse NMF and pruning deep neural network tasks (Section 5).

   - For NMF, GSP competes with state-of-the-art methods on an image dataset and outperforms them on a synthetic dataset (Section 5.1).
   - In pruning deep neural networks, GSP achieves higher accuracies at corresponding sparsity values compared to competing methods on the CIFAR10 dataset. On the Imagenet task, it outperforms a range of pruning methods in terms of sparsity versus accuracy trade-off (Section 5.2).
   - GSP can also recover competitive accuracy with a single projection of large pre-trained models followed by finetuning, altogether skipping the training with regularization phase. (Section 5.3).

Please check the appendix for detailed proofs, implementation details, and additional experiments.

**Notation** We denote $\mathbb{R}^n$ the set of $n$-dimensional vectors in $\mathbb{R}$, $\mathbb{R}_+^n = \mathbb{R}^n \cap \{x \mid x \geq 0\}$ where $x \geq 0$ means that the vector $x$ is component-wise nonnegative, $\mathbb{R}_0^n = \mathbb{R}^n \backslash \{0\}$ and $\mathbb{R}_{0,+}^n = \mathbb{R}_+^n \backslash \{0\}$ where $0$ is the vector of zeros of appropriate dimension. For $x \in \mathbb{R}^n$, we denote $\text{sign}(x)$ the vector of signs of the entries of $x$, $|x|$ the component-wise absolute value of the vector $x$, $[x]_+ = \max(0, x)$ the projection onto the nonnegative orthant, $x(i)$ the $i$th entry of $x$, $\|x\|_0$ the number of nonzero entries of $x$, that is, the $\ell_0$ norm, $\|x\|_1 = \sum_{i=1}^n |x(i)|$ the $\ell_1$ norm of $x$, and $\|x\|_2 = \sqrt{\sum_{i=1}^n x(i)^2}$ the $\ell_2$ norm of $x$. We also denote $\circ$ the component-wise product between two vectors, that is, $z = x \circ y \iff z(i) = x(i)y(i) \; \forall i$, $\mathbf{1}$ the vector of all ones of appropriate dimension, and $[\![1, r]\!] = \{1, 2, \ldots, r\}$.

## 2 Background on Projection with the Hoyer Sparsity Measure

In this section, we define the problem of sparse projection and formulate the sparse projection task for a single vector.

### 2.1 Hoyer sparsity measure

Given a vector $x \in \mathbb{R}^n$, a meaningful way to measure its sparsity is to consider the Hoyer sparsity defined in (1). A main advantage of $\text{sp}(x)$ compared to $\|x\|_0$ is that $\text{sp}(x)$ is smooth except on a set of measure zero, namely when $x(i) = 0$ for some $i$, since $\|x\|_1$ is not smooth at these points. However, it is continuous everywhere except at the origin, $x = 0$, where it is not defined. Moreover, when considering only the nonnegative orthant, that is, $x \in \mathbb{R}_+^n$, then $\text{sp}(x)$ is smooth everywhere except at the origin, since $\|x\|_1 = \mathbf{1}^T x$ for $x \geq 0$. As we will see in Sections 2.2 and 3, see in particular (4) and (GSP), one can restrict the search space to the nonnegative orthant which makes $\text{sp}(x)$ even more convenient to work with. These are crucial properties that allows one to project efficiently onto the set of vectors of a given Hoyer sparsity (Thom et al., 2015). Note that $\text{sp}(x)$ is invariant to scaling, that is, $\text{sp}(x) = \text{sp}(\alpha x)$ for any $\alpha \neq 0$.

Another useful property of $\text{sp}(x)$ is its nonincreasingness under the soft thresholding operator: Given a vector $x$ and a parameter $\lambda \geq 0$, the soft-thresholding operator is defined as

$$\text{st}(x, \lambda) = \text{sign}(x) \circ [|x| - \lambda \mathbf{1}]_+.$$

This property will be particularly useful later when deriving our novel projection of a set of vectors. Note that for $\lambda$ between the largest and second largest entry of $|x|$, $\text{st}(x, \lambda)$ is 1-sparse with $\text{sp}(\text{st}(x, \lambda)) = 1$, hence $\text{sp}(\text{st}(x, \lambda))$ is constant. Interestingly, for $x = b\mathbf{1}$ for some constant $b$, it is not possible to sparsify $x$ (because we cannot differentiate between its entries) and $\text{st}(x, \lambda)$ is constant for all $\lambda < b$. Note that while the $\ell_0$ norm can also be used directly as a sparsity measure (Bolte et al., 2014; Pock & Sabach, 2016), it does not enjoy some of the nice properties of the Hoyer-sparsity measure; see Appendix A.1, Hurley & Rickard (2009) and Thom et al. (2015) for more detailed discussions.

### 2.2 Single Vector Sparse Projection

Let us first present the sparse projection problem for a single vector, along with a reformulation that will be useful to project a set of vectors. These derivations are similar to that of Hoyer (2004); Potluru et al. (2013); Thom et al. (2015). Given $c \in \mathbb{R}_0^n$ and a sparsity level $s \in [0, 1]$, the sparse projection problem can be formulated as follows

$$\min_{z \in \mathbb{R}^n} \|c - z\|_2 \quad \text{such that} \quad \text{sp}(z) \geq s. \tag{3}$$

Let us use the change of variable $z = \alpha x$, with $\alpha = \|z\|_2 \geq 0$ and $\|x\|_2 = 1$. Note that $\text{sp}(z) = \text{sp}(\alpha x) = \text{sp}(x)$ since $\text{sp}(.)$ is invariant to scaling. Hence $\alpha$ does not appear in the sparsity constraints. Moreover, $\alpha$ can be optimized easily. By expanding the $\ell_2$ norm we have:

$$\alpha^* = \text{argmin}_{\alpha \geq 0} \|c - \alpha x\|_2 = \max(0, x^T c),$$

since $\|x\|_2 = 1$. For $\alpha^* > 0$, we have

$$\|c - \alpha^* x\|_2^2 = \|c\|_2^2 - 2\alpha^* x^T c + (\alpha^*)^2 = \|c\|_2^2 - (x^T c)^2.$$

We notice that the sign of the entries of $x$ can be chosen freely since the constraints are not influenced by flipping the sign of entries of $x$. This implies that, at optimality,

- the entries of $x$ will have the same sign as the entries of $c$, and
- $\alpha^* > 0$ since $c \neq 0$ and $x \neq 0$.

Therefore, (3) can be reformulated as

$$
\begin{aligned}
&\max_{x \in \mathbb{R}_0^n} \; x^T |c| \\
&\text{such that} \quad \|x\|_2 = 1, x \geq 0 \text{ and } \operatorname{sp}(x) \geq s.
\end{aligned} \tag{4}
$$

In fact, the optimal solution of (3) is given by $z^* = (|c|^T x^*) \operatorname{sign}(c) \circ x^*$, where $x^*$ is an optimal solution of (4). Although this reformulation is relatively straightforward, it was not present in the literature, as far as we know.

## 3  Grouped Sparse Projection (GSP)

We extend the argument of Section 2.2 to a set of vectors and present our novel approach to grouped sparse projection.

### 3.1  Formulation of the Problem

Let $\{c_i \in \mathbb{R}_0^{n_i}\}_{i=1}^r$ be a set of non-zero vectors. The main goal of sparse projection is to find a set of non-zero unit-norm vectors $\{x_i \in \mathbb{R}_0^{n_i}\}_{i=1}^r$ that has an average target sparsity larger than a given $s \in [0, 1]$. Mathematically, let $\boldsymbol{\mathcal{X}} = \{x_i \in \mathbb{R}_0^{n_i}, i \in [\![1, r]\!] : x_i \geq 0, \|x_i\|_2 = 1 \; \forall i\}$. We propose the following novel *grouped sparse projection* problem:

$$
\max_{\boldsymbol{\mathcal{X}}} \; \sum_{i=1}^r x_i^T |c_i| \text{ such that } \frac{1}{r} \sum_{i=1}^r \operatorname{sp}(x_i) \geq s. \tag{GSP}
$$

The main reason for the choice of this formulation is that it makes GSP much faster to solve. In fact, as for (3) that was studied by Thom et al. (2015), we will be able to reduce this problem to the root finding problem of a nonincreasing function in one variable. In particular, using the objective function $\min_{\boldsymbol{\mathcal{X}}} \sum_{i=1}^r \|c_i - x_i\|_2$ (or $\sum_{i=1}^r \|c_i - x_i\|_2^2$) would not allow such an effective optimization scheme.

*Remark* 3.1 (Abuse of terminology). The solution to the problem (GSP) is not projection, as it does not provide a point within a set closest in some norm to a given point. However, for simplicity, and since it extends the projection in the case of a single vector, see (3) and (4), we abuse the terminology and refer to (GSP) as a projection.

Let us reformulate GSP focusing on the sparsity constraint: we have

$$
\sum_{i=1}^r \operatorname{sp}(x_i) = \sum_{i=1}^r \frac{\sqrt{n_i} - \|x_i\|_1}{\sqrt{n_i} - 1} = \sum_{i=1}^r \frac{\sqrt{n_i}}{\sqrt{n_i} - 1} - \sum_{i=1}^r \frac{\|x_i\|_1}{\sqrt{n_i} - 1} \geq rs,
$$

where we used the fact that $\|x_i\|_2 = 1$ for all $i$ in the feasible set $\boldsymbol{\mathcal{X}}$. Denoting $k_s = \sum_{i=1}^r \frac{\sqrt{n_i}}{\sqrt{n_i} - 1} - rs$ and $\beta_i = \frac{1}{\sqrt{n_i} - 1}$, GSP can be reformulated as follows

$$
\max_{\boldsymbol{\mathcal{X}}} \sum_{i=1}^r x_i^T |c_i| \quad \text{such that} \quad \sum_{i=1}^r \beta_i \mathbf{1}^T x_i \leq k_s. \tag{5}
$$

We used $\|x_i\|_1 = \mathbf{1}^T x_i$ since $x_i \geq 0$. Note that the maximum of (5) is attained since the objective function is continuous and the feasible set is compact (extreme value theorem).

## 3.2 Lagrange Dual Formulation

In order to solve (5), we follow a standard dual approach, similarly as done by Thom et al. (2015) for the projection of a single vector. However, our derivations are rather different because the vectors to be projected share the same Lagrange dual variable, while we need to carefully treat the case when some of the vectors are projected onto a 1-sparse vector. Let us introduce the Lagrange variable $\mu \geq 0$ associated with the constraint $\sum_{i=1}^{r} \beta_i \mathbf{1}^T x_i \leq k_s$. The Lagrange dual function with respect to $\mu$ is given by

$$
\begin{aligned}
\ell(\mu) &= \max_{\mathcal{X}} \sum_{i=1}^{r} x_i^T |c_i| - \mu \left( \sum_{i=1}^{r} \beta_i x_i^T \mathbf{1} - k_s \right) \\
&= \max_{\mathcal{X}} \sum_{i=1}^{r} x_i^T (|c_i| - \beta_i \mu \mathbf{1}) + \mu k_s.
\end{aligned}
\tag{6}
$$

The dual problem is given by $\min_{\mu \geq 0} \ell(\mu)$. The optimization problem to be solved to compute $\ell(\mu)$ is separable in variables $x_i$'s and consequently can be solved individually for each $x_i$. Let us denote $x_i[\mu]$ the optimal solution of (6). For each $i$, there are two possible cases, depending on the value of $\mu$:

1. $|c_i| - \mu \beta_i \mathbf{1} > 0$: the optimal $x_i[\mu]$ is given by

$$
x_i[\mu] = \frac{[|c_i| - \mu \beta_i \mathbf{1}]_+}{\left\| [|c_i| - \mu \beta_i \mathbf{1}]_+ \right\|_2} = \frac{\operatorname{st}(|c_i|, \mu \beta_i)}{\left\| \operatorname{st}(|c_i|, \mu \beta_i) \right\|_2}.
$$

   This formula can be derived from the first-order optimality conditions. Note that this formula is similar to that of Thom et al. (2015). The difference is that the $x_i$'s share the same Lagrange variable $\mu$.

2. $|c_i| - \mu \beta_i \mathbf{1} \leq 0$: the optimal $x_i[\mu]$ is given by the 1-sparse vector whose nonzero entry corresponds to the largest entry of $|c_i| - \mu \beta_i \mathbf{1}$, that is, of $|c_i|$. Note that if the largest entry of $|c_i|$ is attained for several indexes, then the optimal 1-sparse solution $x_i^*$ is not unique. Note also that this case coincides with the case above for $\mu$ in the interval between the largest and second largest entry of $c_i$.

## 3.3 Characterizing the Optimal Solution $x_i[\mu]$

Let $\tilde{\mu}$ be the smallest value of $\mu$ such that $x_i[\tilde{\mu}]$ are all 1-sparse. There are three scenarios based on the value of $\mu$: (a) $\mu = 0$, (b) $\mu \geq \tilde{\mu}$ and (c) $0 < \mu < \tilde{\mu}$.

(a) For $\mu = 0$, we have $x_i[0] = \frac{|c_i|}{\|c_i\|_2}$. If $x_i[0]$ is feasible, that is, $\sum_{i=1}^{r} \beta_i \mathbf{1}^T x_i[0] \leq k_s$, then it is optimal since the error of GSP is zero: this happens when the $c_i$'s are already sparse enough and do not need to be projected.

(b) For $\mu \geq \tilde{\mu}$, all $x_i[\tilde{\mu}]$ are all 1-sparse so that $\mathbf{1}^T x_i[\tilde{\mu}] = 1$ hence

$$
\begin{aligned}
g(\tilde{\mu}) &= \sum_{i=1}^{r} \beta_i - k_s \\
&= \sum_{i=1}^{r} \frac{1}{\sqrt{n_i} - 1} - \sum_{i=1}^{r} \frac{\sqrt{n_i}}{\sqrt{n_i} - 1} + rs \\
&= r(s - 1) \leq 0.
\end{aligned}
\tag{7}
$$

   The value $\tilde{\mu}$ is given by the second largest entry among the vectors $|c_i|/\beta_i$'s. In fact, if $\mu$ is larger than the second largest entry of $|c_i|$, then $x_i[\mu]$ is 1-sparse. Note that if the largest and second largest entry of a vector $|c_i|$ are equal, then $g(\mu)$ is discontinuous. This is an unavoidable issue when one wants to make a vector sparse: if the largest entries are equal to one another, one has to decide which one will be set to zero. (For example, $[1, 0]$ and $[0, 1]$ are equally good 1-sparse representation of $[1, 1]$.)

---

**Algorithm 1** GSP($\{c_i \in \mathbb{R}^{n_i}\}_{i=1}^r, s, \epsilon, r_l$)

---

1: **input:** $\{c_i \in \mathbb{R}^{n_i}\}_{i=1}^r$, the average sparsity $s \in [0, 1]$, the accuracy $\epsilon$, the parameter $r_l \in [1/2, 1)$
2: **output:** $\{z_i \in \mathbb{R}^{n_i}\}_{i=1}^r$ with average sparsity in $[s - \epsilon, s + \epsilon]$
3: $\underline{\mu} = 0$, $\bar{\mu} = \tilde{\mu}$, $\mu^* = 0$, $\Delta = \bar{\mu} - \underline{\mu}$.
4: **while** $|g(\mu^*)| > r\epsilon$ **do**
5:     $\mu^{old} = \mu^*$
6:     $\mu^* = \mu^* + \frac{k - g(\mu^*)}{g'(\mu^*)}$                             ▷ Newton's step
7:     **if** $\mu^* \notin [\underline{\mu}, \bar{\mu}]$ **then** $\mu^* = \frac{\underline{\mu} + \bar{\mu}}{2}$ **end if**      ▷ Bisection method if Newton's step fails
8:     **if** $g(\mu^*) > 0$ **then** $\underline{\mu} = \mu^*$ **else** $\bar{\mu} = \mu^*$ **end if**      ▷ Update feasible interval
9:     **if** $\bar{\mu} - \underline{\mu} > r_l \Delta$ and $|\mu^{old} - \mu^*| < (1 - r_l)\Delta$ **then**
10:         $\mu^* = \frac{\underline{\mu} + \bar{\mu}}{2}$                  ▷ If feasible interval is not reduced use bisection again
11:         **if** $g(\mu^*) > 0$ **then** $\underline{\mu} = \mu^*$ **else** $\bar{\mu} = \mu^*$ **end if**      ▷ Update feasible interval
12:     **end if**
13:     $\Delta = \bar{\mu} - \underline{\mu}$                                  ▷ Update diameter $\Delta$
14:     **if** $\Delta < \epsilon\mu^*$ and $|\mu^* - \mu| < \epsilon\mu^*$ for some $\mu \in \mathcal{D}$ **then**
15:         break;                ▷ $\mathcal{D}$ contains the set of discontinuous points, stop GSP
16:     **end if**
17: **end while**
18: **return** $z_i^* = \{|c_i|^T x_i[\mu^*] \operatorname{sign}(c_i) \circ x_i[\mu^*]\}_{i=1}^r$ ▷ Compute $x_i[\mu^*]$ (see Section 3.2) and return projections

---

(c) For $0 < \mu < \tilde{\mu}$, the constraint is active at optimality and we need to find the value $\mu$ such that

$$g(\mu) = \sum_{i=1}^r \beta_i \mathbf{1}^T x_i[\mu] - k_s = 0,$$

that is, find a root of $g(\mu)$. Theorem 3.2 guarantees that the solution $\mu^*$ is unique in this setting, and Corollary 3.3 proves the respective projection $x[\mu^*] = [x_1[\mu^*], \ldots, x_r[\mu^*]]$ is also unique.

**Theorem 3.2** (Uniqueness of $\mu^*$). *The function $g(\mu)$ is strictly decreasing for $0 < \mu < \tilde{\mu}$. Hence, it is not discontinuous around $g(\mu) = 0$ and attains a unique root $\mu^*$.*

*Proof sketch.* The function $g(\mu)$ is continuous as it is a continuous function of $x(\mu)$, which is continuous. Hence, it suffices to show that $g'(\mu) < 0$ for $0 < \mu < \tilde{\mu}$. See Appendix C.3 for a detailed derivation in a more general case. □

**Corollary 3.3** (Uniqueness of projection $x^*$). *If the largest entry of each $|c_i|$ is attained for only one entry, the projection $x[\mu^*]$ is unique.*

*Proof sketch.* If $\mu = 0$ and $\mu > \tilde{\mu}$, $x(\mu)$ is unique provided the largest and second entry of each $|c_i|$ are not equal (see Section 3.3). For $0 < \mu < \tilde{\mu}$, since the optimal $\mu^*$ is unique it suffices to prove $x_i[\mu]$ is a one-to-one mapping, i.e., that if $x_i(\mu_1) = x_i(\mu_2)$ then $\mu_1 = \mu_2$. Detailed derivations shows that $x_i[\mu]$ is a one-to-one mapping unless all entries of $c_i$ are equal, which is in contrast with the assumption (see Appendix B). □

Once the optimal $\mu^*$ and the respective projections $x_i[\mu^*]$ are computed, one can retrieve the sparse projections via $z_i^* = (|c_i|^T x_i[\mu^*]) \operatorname{sign}(c_i) \circ x_i[\mu^*]$ for $i \in [\![1, r]\!]$.

## 3.4 Implementation and Computational Cost

In order to compute the dual variable $\mu$, which leads to the desired average Hoyer sparsity for the vectors $x_i[\mu]$'s, we resort to a standard approach to compute the root of the nonlinear function, $g(\mu)$, namely Newton's method which has quadratic convergence (given that we are sufficiently close to a root, and that the function is sufficiently smooth). Because $g(\mu)$ is not smooth everywhere, we have implemented the bisection method

as a fallback procedure, which guarantees the algorithm's computational complexity to be upper-bounded by $\mathcal{O}(N \log N)$. The use of bisection is necessary if Newton's method either goes outside of the current feasible interval $[\underline{\mu}, \bar{\mu}]$ containing the solution $\mu^*$ or stagnates locally and does not converge (although we have not observed the latter behavior in practice). This is a a standard textbook strategy in numerical analysis.

Algorithm 1 summarizes the GSP algorithm. The main computational cost of GSP is to compute $g(\mu)$ and $g'(\mu)$ at each iteration, which requires $\mathcal{O}(N)$ operations where $N = \sum_{i=1}^{r} n_i$. Most of the computational cost resides in computing the $x_i[\mu]$'s and some inner products. The total computational cost is $\mathcal{O}(tN)$, where $t$ is the number of Newton's method iterations, hence making GSP linear in the size of the problem. In practice, we observed that Netwon's method converges very fast and does not require many iterations $t$ (see Appendix A.4 for a synthetic data example with vectors of dimension $n_i = 1000$, in which Newton's achieves convergence in $t = 4$ iterations or less). In our experiments we have observed that using Newton's method with initial point $\mu = 0$ performs well, which is in line with the results of Thom et al. (2015), as points where the function is not differentiable typically form a set of measure zero (Kummer et al., 1988). The reason to choose $\mu = 0$ as the initial point is because $g(\mu)$ decreases initially fast (all entries of $x_i$ corresponding to a nonzero entry of $c_i$ are decreasing) while it tends to saturate for a larger $\mu$. In particular, we cannot initialize $\mu$ at values larger than $\tilde{\mu}$ since $g(\mu)$ is constant ($g'(\mu) = 0$) for all $\mu \geq \tilde{\mu}$. Finally, we have included a method to detect discontinuities, corresponding to situations when the largest entries of the $c_i$'s are not unique. In that case we require GSP to return a $\mu$ such that $g(\mu + \epsilon\mu) < 0 < g(\mu - \epsilon\mu)$, where $\epsilon$ is a desired accuracy (which in practice can be set to $\epsilon = 10^{-4}$).

## 4 Weighted Grouped Sparse Projection

In some settings, one may want to minimize a weighted sum of the entries of the $x_i$'s, that is, $\sum_{i=1}^{r} w_i^T x_i \leq k$ for some $w_i \in \mathbb{R}_+^{n_i}$. This amounts to replacing the $\ell_1$-norm in the sparsity measure sp(.) by a weighted $\ell_1$ norm. We introduce the notion of *weighted sparsity*: For a vector $x \in \mathbb{R}_0^n$ and given $w \in \mathbb{R}_{0,+}^n$, we define

$$\text{sp}_w(x) = \frac{\|w\|_2 - \frac{\|Wx\|_1}{\|x\|_2}}{\|w\|_2 - \min_i w(i)} \in [0,1], \tag{8}$$

where $W = \text{diag}(w)$. We use the weighted sparsity in (8) to define the $w$-sparse projection problem of the set of vectors $\{c_i \in \mathbb{R}^{n_i}\}_{i=1}^{r}$. Let $\{w_i \in \mathbb{R}_{0,+}^{n_i}\}_{i=1}^{r}$ be the nonzero nonnegative weight vectors associated with the $c_i$'s. Given a target average weighted sparsity $s_w \in [0,1]$, we formulate the weighted group sparse projection (WGSP) problem as follows:

$$\max_{\mathcal{X}} \sum_{i=1}^{r} x_i^T |c_i| \quad \text{such that} \quad \frac{1}{r} \sum_{i=1}^{r} \text{sp}_{w_i}(x_i) \geq s_w. \tag{WGSP}$$

We have included properties of the proposed weighted sparsity, along with full derivation of the WGSP problem, the optimal solutions and experiments in Appendix C.

## 5 Experiments

In this section, we evaluate the utility of GSP in two applications: sparse NMF and deep learning network pruning. The goal is to evaluate the effect of performance at desired sparsity in the unsupervised and supervised setting.

### 5.1 gspNMF: Projection-based Sparse NMF

As explained in Section 1.1, the standard NMF problem is formulated as in (2). As studied in detail in the recent book by Gillis (2020), the most widely used and efficient algorithms for NMF are block coordinate descent algorithms, and two of the most popular and effective ones are the following:

1. A-HALS updates alternatively the columns of X and the rows of H using a closed-form solution (Gillis & Glineur, 2012),

2. NeNMF updates alternatively the full factors X and H. The update of each factor requires to solve convex non-negative least squares problems that NeNMF solves approximately using a few steps of Nesterov fast gradient method (FGM), an optimal first-order method (Guan et al., 2012).

Sparse NMF enforces an additional sparsity constraint on X and our GSP leads to a natural sparse NMF formulation: given an average sparsity level for the columns of $X$, $s$, solve

$$\min_{X\in\mathbb{R}^{m\times r}, H\in\mathbb{R}^{r\times n}} \|Y - XH\|_F^2$$
$$\text{such that } X \geq 0, H \geq 0 \text{ and } \frac{1}{r}\sum_{k=1}^{r} \text{sp}(X(:,k)) \geq s, \tag{9}$$

where $X(:,k)$ is the $k$th column of X. To solve (9), we use the same update as A-HALS for H as it is not affected by the sparsity requirement. For updating X, we adapt the update of NeNMF (Guan et al., 2012) by replacing the projection onto the nonnegative orthant with GSP; we refer to this new sparse NMF algorithm as group sparse projection NMF (gspNMF). Note that our feasible set is not convex, hence FGM is not guaranteed to converge. However, we use a restarting scheme and keep the best iterate in memory. We compare gspNMF with

1. The same algorithm where the projection is performed column-wise with the columns constrained to have the same sparsity level, as done by Thom et al. (2015). This algorithm solves the same problem (9) where the last constraint is replaced by $\text{sp}(X(:,k)) \geq s$ for all $k$. We refer to this algorithm as cspNMF.

2. A-HALS with $\ell_1$ penalty described by Gillis (2012) that uses the formulation

$$\min_{X\in\mathbb{R}^{m\times r}, H\in\mathbb{R}^{r\times n}} \|Y - XH\|_F^2 + \sum_{k=1}^{k} \lambda_k \|X(:,k)\|_1$$
$$\text{such that } X \geq 0, H \geq 0, \text{ and } \|X(:,k)\|_\infty = 1 \text{ for } k \in [\![1,r]\!], \tag{10}$$

where the $\ell_1$ penalty parameters for each column of X, $\lambda_k$'s, are automatically tuned to achieve the same desired $\ell_0$ sparsity level[1]; see the code provided by the author. We refer to this algorithm as $\ell_1$ A-HALS.

We compare all algorithms in terms of error per iteration. Since all algorithms have almost the same computational complexity per iteration[2], we believe it is fair and more insightful to compare these five algorithms with respect to the iteration number. In all experiments, we perform 500 iterations (updates of X and H) of each algorithm. See Appendix D for details about the experimental parameters and datasets used.

**gspNMF on synthetic data**  We first perform experiments on synthetic data sets where W and H are generated randomly, with W having about 50% of its entries equal to zero, and with setting $m = n = 100$ and $r = 10$; see Appendix D.1 for the details. We note an interesting finding: if gspNMF is given the sparsity of an exact factorization, it converges to an exact solution much faster than A-HALS and NeNMF. In other words, gspNMF is able to use the prior information to its advantage. Although A-HALS and NeNMF are less constrained, as they solve (2) instead of (9), they converge significantly slower. From Figure 1 we observe

---

[1]Although $\ell_1$ A-HALS does not use the same sparsity measure as gspNMF and cspNMF, these two measures are close to one another in most cases. For example, in the synthetic data experiments (see below), we will generate $n$-dimensional vectors using the Gaussian distribution and setting the negative entries to zero. Such vectors have an expected $\ell_0$ sparsity of 50%. Generating 100 such vectors, their average Hoyer sparsity is 48.31% while their average $\ell_0$ sparsity is 49.9%.

[2]The main computation cost resides in computing $YH^T$ (resp. YX) when updating X (resp. H) which requires $O(mnr)$ operations. gspNMF and cspNMF are slightly more expensive because of the projection step, although this is not the main computational cost (the projection cost is linear in $m$ and $r$). For example, on synthetic data sets for $n$ sufficiently large (see below), gspNMF and cspNMF took on average the same computational time than NeNMF and A-HALS (in fact, for some runs with $n = 10^4$, gspNMF surprisingly ran slightly faster than NeNMF and A-HALS, due to inaccuracy in accounting for the computational time).

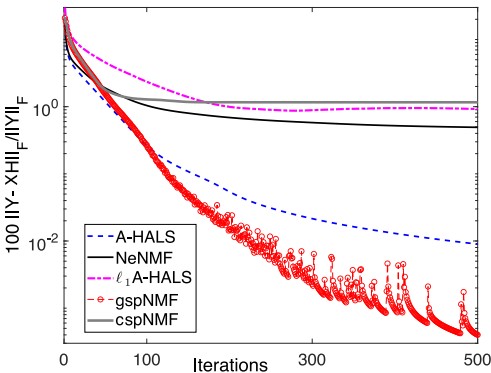 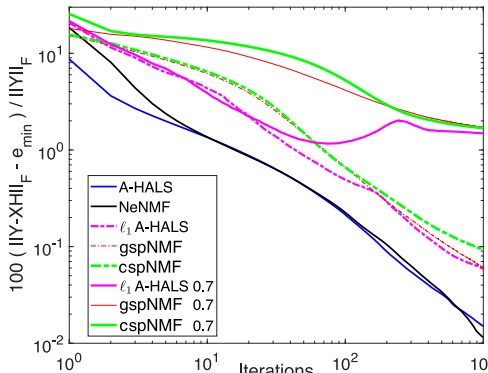

Figure 1: Comparison of NMF and sparse NMF algorithms. On the left: Average relative error $100\frac{\|Y-XH\|_F - e_{\min}}{\|Y\|_F}$ obtained with different NMF algorithms over 50 synthetic data sets. On the right: Average relative error in percent to which the lowest error $e_{\min}$ obtained among all algorithms and all initializations is subtracted (hence the error should go to zero for the best algorithm), that is, $100\frac{\|Y-XH\|_F - e_{\min}}{\|Y\|_F}$, over 10 random initializations on the CBCL data set with $r = 49$.

that gspNMF performs better than A-HALS, which in turn outperforms NeNMF. gspNMF reduces the error towards zero much faster, although having a higher initial error as it is more constrained.

Since cspNMF is more constrained, it is not able to perform as well as gspNMF. This illustrates the significant benefit of using an average sparsity instead of a single sparsity level for all columns of X. For similar reasons, $\ell_1$ A-HALS performs worse as it attempts to achieve the same given sparsity level for all columns of X.

**gspNMF on CBCL facial image data** We test our method on one of the most widely used datasets in the NMF literature, the CBCL facial images used in the seminal work by Lee & Seung (1999) with $r = 49$. NeNMF and A-HALS for NMF (2) generate solution with average Hoyer sparsity about 70%. We run the sparse NMF techniques with sparsity set at 85%, hence we expect sparse NMF to have higher approximation errors but have higher sparsity. Using 10 random initializations, Figure 1 reports the evolution of the average relative error. In this case, gspNMF performs similar to the state-of-the-art algorithms, cspNMF and $\ell_1$ A-HALS, because the columns of X have similar sparsity levels. We include the basis elements obtained by each different methods in Fig 5 of Appendix-D.2.

## 5.2 Pruning Deep Neural Networks with GSP

We evaluate performance of GSP in pruning each layer of a modern convolutional neural network using a fixed value of the sparsity parameter $s$. Although we chose to prune each layer with the same sparsity to simplify the comparison, GSP can easily be used with different sparsity levels for parameters of different layers, providing fine control over the sparsity of the network.

To simplify comparison with related work we express the sparsity of the pruned network in terms of the number of zeroed parameters instead of $\text{sp}(x)$ (1). Since $\text{sp}(x)$ is a differentiable approximation of the $\ell_0$ norm, applying GSP with sparsity $s$ to a layer of the network pushes $s$ fraction of the parameters to zero or near zero, thus retaining interpretability of the parameter $s$.

We run experiments on the CIFAR-10 (Krizhevsky et al., 2009) dataset with the VGG16 model and on the ILSVRC2012 Imagenet dataset (Russakovsky et al., 2015) with the ResNet50 model (He et al., 2016)[3]. For the fully connected layers, we project the connections in each layer separately, with a target sparsity $s$. This ensures the weights of that particular layer have $\text{sp}(x) = s$. For the convolutional layers, we project each $c \times k \times k$ filter treating each as a vector $x_i \in \mathbb{R}^n$ in our formulation, where $n = c \times k^2$ and $c$ is the number of input channels. All the filters in a particular layer are projected at once.

---

[3]The code is available at https://github.com/riohib/gsp-for-deeplearning

| VGG16 on CIFAR-10 | Model Sparsity | | | | |
|---|---|---|---|---|---|
| Methods | 80% | 85% | 90% | 95% | 97% |
| Dense Baseline | 92.82 % | - | - | - | - |
| Random Pruning | 83.7% | 82.75% | 81.56 % | 78.18% | 76.3% |
| Magnitude Pruning (Han et al., 2015) | **92.78%** | **92.74%** | **92.5%** | 91.41% | 10% |
| DeepHoyer (Yang et al., 2020) | 91.23% | 91.2% | 91.42% | 91.47% | 91.54% |
| GSP | 92.37% | 92.28% | 92.39% | **92.32%** | **91.92%** |

Table 1: Test Accuracy of pruned VGG16 network using GSP vs random pruning and other pruning methods for sparsity varying in the 80% to 97% range.

We test two pruning strategies using GSP. The first strategy consists in projecting the layers of a model every specified number of iterations, then pruning the weights post-training followed by finetuning of the surviving weights – we call this "induced-GSP" training. In the second strategy, we start with a pretrained model, project the layers once, and finetune the model. The second approach skips the sparsity inducing training phase, which is a requirement for popular regularization-based sparse techniques. We denote this approach as "single-shot GSP".

Table 1 reports the test accuracy of induced-GSP at different sparsity levels against (a) random pruning, which randomly selects $(1-s)$ fraction of the model weights and retrains them, (b) magnitude pruning, where the top $(1-s)$ fraction of the parameters are selected post-training and retrained and (c) DeepHoyer (DH, Yang et al. 2020), a high-performance regularization-based pruning technique. GSP clearly outperforms random pruning and finds out important connections in the network. While magnitude pruning performs slightly better than GSP at lower sparsity levels, GSP achieves comparable and higher accuracies in the sparsity region $s > 90\%$. Finally, GSP achieves higher accuracy than DH throughout the different sparsity levels tested. We note that since DH requires a regularization parameter to be tweaked using naive search, we trained upwards of 60 models to achieve the reported performance. This quickly becomes infeasible for training ImageNet level tasks without using significant resources.

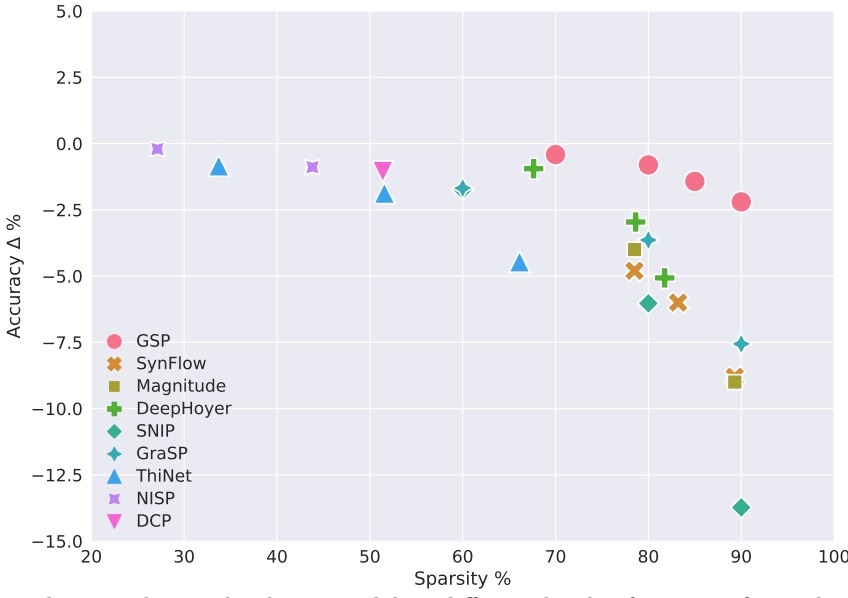

Figure 2: Accuracy change relative the dense model at different levels of sparsity for multiple state-of-the-art network pruning techniques applied to ResNet50 on the ImageNet dataset. Values towards top-right are better. Notably, GSP provides superior sparsity vs accuracy trade-off.

**ResNet50 on ImageNet** Testing pruning techniques on the ImageNet dataset (Russakovsky et al., 2015) is becoming a standard in the field of model compression as it is a large-scale complex dataset. Therefore, we test the efficacy of GSP on the ImageNet dataset and compare it with a range of pruning techniques from the literature. We compare against regularization-based (Yang et al., 2020), magnitude-based (Han et al., 2016), and structure-based pruning (Luo et al., 2017; Yu et al., 2018; Zhuang et al., 2018). We also test GSP against pruning at initialization techniques by Lee et al. (2019); Wang et al. (2020); Tanaka et al. (2020), that have the added benefit of training a sparse model but the disadvantage of having almost no training information to guide pruning. We project the layers of a Res-Net50 model with GSP every fixed iterations and subsequently prune the parameters identified by our method according to set sparsity values. Pruning a network is a trade-off between sparsity and accuracy. Hence, in Figure 2 we report the sparsity and accuracy of GSP against other pruning techniques. We observe that even at relatively high sparsity levels, GSP competes with and outperforms all pruning techniques we compared with, dropping only $0.41, 0.8, 1.43$ and $2.2$ percentage points at $70, 80, 85$ and $90$ percent sparsity respectively.

### 5.3 Single Shot Network Pruning

GSP can also be utilized to generate sparse models by a single projection step instead of repeated projections throughout the training process. Thus, we can first take a pretrained model, project the model once layerwise with our choice of sparsity, and finally finetune the surviving connections. We refer to this approach as "Single-Shot GSP". This is in contrast to popular regularization-based techniques of Yang et al. (2020) and Ma et al. (2019) which need to train with the regularizer first, then prune the weights and finally finetune the surviving weights to generate the final sparse model. The ability to project a pretrained model directly without going through a regularized training phase would be an attractive ability to have, e.g. in transfer learning.

Table 2 shows the accuracy and sparsity value of induced GSP and single-shot GSP against DeepHoyer (DH) on the ResNet-56 and ResNet-110 architectures and tested on the CIFAR-10 Dataset. The drop in accuracy compared to their respective baselines is reported as well. Noticeably, single-shot GSP performs comparably or better than DH. Particularly, we notice that for the ResNet-56 architecture, single-shot GSP achieves a higher sparsity of 85.75% with the same accuracy drop of 0.42% compared to DH. For the ResNet-110 architechture, we find that even with a higher sparsity of 85.90% we achieve lower accuracy drop compared to DH. Subsequently, even after pushing the single-shot GSP sparsity to 90.72%, the accuracy drop was only 1.17%, still lower than DH's 1.24% at 83.94% sparsity. This result is even more salient as DH relies on an extra sparsity-inducing regularizer phase with 164 epochs of training to first induce sparsity, and then prunes the model with a threshold followed by finetuning for similar number of epochs. In contrast, we use a preset sparsity value for GSP, project a pretrained model once, and finetune the surviving weights.

**Summary of the numerical experiments** We have shown that GSP in the NMF application performs competitively with state-of-the-art approaches on image data sets without the need to tune the sparsity parameters and outperforms them on synthetic and sparse data sets. In deep network pruning, the use of GSP leads to sparser and more accurate networks without extra training time needed to tune sparsity parameters. On ImageNet data with ResNet50 model GSP outperforms a large number of existing approaches. It also demonstrates superior performance when used for single-shot pruning.

## 6 Conclusion

We proposed a novel grouped sparse projection algorithm which allows us to project a set of vectors to a set of sparse vectors whose average sparsity is controlled via a single interpretable parameter. This parameter should be carefully chosen by the user to obtain a good tradeoff between the sparsity of the projected vectors and how well they approximate the original input vectors; see, e.g., Table 1 and Figure 2 in the context of pruning deep neural networks. We provided theoretical guarantees for the uniqueness of the solution of the dual problem which is also shown to correspond to the uniqueness of the primal solution. Additionally, we proposed an efficient algorithm whose running time is shown to be linear in the size of the problem. We demonstrate the efficacy of the projection in two important learning applications, namely in pruning deep

| Methods | Architecture | Sparsity% | Accuracy% | Drop % |
|---|---|---|---|---|
| Induced GSP | ResNet-56 | 88.33 | 92.04 | 1.09 |
| DH | ResNet-56 | 84.64 | 92.71 | 0.42 |
| **Single-shot GSP** | ResNet-56 | **85.78** | **92.71** | **0.42** |
| Induced GSP | Resnet-110 | 95.18 | 92.67 | 0.95 |
| DH | ResNet-110 | 83.94 | 92.76 | 1.24 |
| **Single-Shot GSP** | ResNet-110 | **85.90** | **92.86** | **0.76** |
| | ResNet-110 | **90.72** | **92.45** | **1.17** |

Table 2: Sparsity vs drop in accuracy for ResNet-56 and ResNet-110 for single-shot and induced GSP (our algorithms) in comparison to DeepHoyer (DH) in the CIFAR-10 dataset. GSP with a single projection produces a model with comparable or better sparsity vs accuracy trade-off than a full pipeline of DeepHoyer's regularized training and finetuning. The best sparsity-accuracy trade-off compared to DH are reported in bold.

neural networks and sparse non-negative matrix factorization and validate it on a wide variety of datasets. However, our projection is not limited to the listed applications and could also be used in dictionary learning, sparse PCA, and in different types of neural network architectures. We expect the proposed group sparse projection operator to have wide applicability given the interpretability of the sparsity measure, practical efficiency of the projection, and strong theoretical guarantees.

## Acknowledgement

We thank the action editor and the three reviewers for their insightful comments that helped us improve the paper significantly.

This study was in part supported by NIH MH129047, NIH DA040487, NIH MH121885 and NSF 2112455.

Nicolas Gillis acknowledges the support by the Fonds de la Recherche Scientifique - FNRS and the Fonds Wetenschappelijk Onderzoek - Vlanderen (FWO) under EOS Project no O005318F-RG47, and by the Francqui Foundation.

Niccolò Dalmasso, Sameena Shah and Vamsi K. Potluru at the time of publishing are at JP Morgan AI Research. This paper was prepared for information purposes by the AI Research Group of JPMorgan Chase & Co and its affiliates ("J.P. Morgan"), and is not a product of the Research Department of J.P. Morgan. J.P. Morgan makes no explicit or implied representation and warranty and accepts no liability, for the completeness, accuracy or reliability of information, or the legal, compliance, financial, tax or accounting effects of matters contained herein. This document is not intended as investment research or investment advice, or a recommendation, offer or solicitation for the purchase or sale of any security, financial instrument, financial product or service, or to be used in any way for evaluating the merits of participating in any transaction, and shall not constitute a solicitation under any jurisdiction or to any person, if such solicitation under such jurisdiction or to such person would be unlawful.

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

# Appendices

## A    Hoyer Sparsity Measure and Grouped Sparse Projection

### A.1    The Hoyer Sparsity Measure

In section 2.1 we discussed the *Hoyer sparsity* measure (Hoyer, 2004) for a given vector $x \in \mathbb{R}^n$ and for $x \neq 0$ and $n > 1$, we defined the sparsity of $x$ as

$$\mathrm{sp}(x) = \frac{\sqrt{n} - \frac{\|x\|_1}{\|x\|_2}}{\sqrt{n} - 1} \; \in \; [0, 1], \tag{11}$$

The above measure has the following properties, which we list here for completeness:

- We have that $\mathrm{sp}(x) = 0 \iff \|x\|_1 = \sqrt{n}\|x\|_2 \iff x(i) = b$ for all $i$ and for some constant $b$, while $\mathrm{sp}(x) = 1 \iff \|x\|_0 = 1$, where $\|x\|_0$ counts the number of nonzero entries of $x$.

- One of the main advantage of $\mathrm{sp}(x)$ compared to $\|x\|_0$ is that $\mathrm{sp}(x)$ is smooth over $\mathbb{R}^n$ apart from a measure-zero set, i.e., when $x = 0$ or $x(i) = 0$ for any $i$. For example, with this measure, the vector $[1, 10^{-6}, 10^{-6}]$ is sparser than $[1, 1, 0]$, which makes sense numerically as $[1, 10^{-6}, 10^{-6}]$ is very close to the 1-sparse vector $[1, 0, 0]$.

- $\mathrm{sp}(x)$ is invariant to scaling, that is, $\mathrm{sp}(x) = \mathrm{sp}(\alpha x)$ for any $\alpha \neq 0$.

- Note that for any two vectors $w$ and $z$, $\mathrm{sp}(w) \leq \mathrm{sp}(z) \iff \frac{\|w\|_1}{\|w\|_2} \geq \frac{\|z\|_1}{\|z\|_2}$.

- Note also that $\mathrm{sp}(x)$ is not defined at 0, nor for $n = 1$.

- It is non-increasing under the soft thresholding operator: Given a vector $x$ and a parameter $\lambda \geq 0$, it is defined as $\mathrm{st}(x, \lambda) = \mathrm{sign}(x) \circ [|x| - \lambda \mathbf{1}]_+$ where $\circ$ is the component-wise multiplication, and $[.]_+$ is the projection onto the non-negative orthant, that is, $\max(0, .)$.

### A.2    Comparison to typical formulations

We discuss how the GSP formulation compares with two typical approaches to sparsify a set of vectors:

- The most popular method to make a set of vectors sparse is arguably to use $\ell_1$ penalty terms, solving

$$\min_{x_i \in \mathbb{R}_0^{n_i}, 1 \leq i \leq r} \; \sum_{i=1}^{r} \frac{1}{2}\|c_i - x_i\|_2^2 + \lambda_i \|x_i\|_1,$$

  for which the solution is given by the soft thresholding operator, $x_i^* = \mathrm{st}(c_i, \lambda_i) \; 1 \leq i \leq r$. This is widely used in algorithms for compressed sensing and for solving inverse problems (in particular, to find sparse solutions to an underdetermined linear system); see, e.g., Beck & Teboulle (2009). The use of $\ell_1$ penalty to obtain sparse factors in low-rank matrix approximations is also arguably the most popular approach; see, e.g., Kim & Park (2007); Journée et al. (2010).

  The main drawback of this approach is that the parameters $\lambda_i$ needs to be tuned to obtain a desired sparsity level, As we noted, our projection resolves this drawback.

- Using the method of Thom et al. (2015) that projects a vector onto a vector with a desired level of sparsity, that is, solves (3), we could either project each vector $c_i$ independently but then we would have to choose a priori the sparsity level of each projection $x_i$. We could also project the single vector $[c_1; c_2; \ldots; c_r] \in \mathbb{R}^{\sum_{i=1}^{r} n_i}$, stacking the $c_i$'s on top of one another. However, some vectors $c_i$

could be projected onto zero, which is not desirable in some applications; for example, in low-rank matrix approximation problems this would make the matrix with $x_i$'s as its columns rank deficient. More pertinently, in the scenario of learning sparse deep models, stacking all the layers together and projecting them this way would result in the projection of all the parameters of a layer onto zero, also known as *layer-collapse* (Tanaka et al., 2020), which makes the model untrainable.

### A.3 Properties of $g(\mu)$ and edge cases

In section 3 we discussed that the optimization problem to be solved to compute $\ell(\mu)$ in (6) is separable in the variables $x_i's$, and that it can be solved individually for each $x_i$. We denoted the optimal solution of the (6) with $x_i[\mu]$ and noted that there are two possible cases, depending on the value of $\mu$. We noted that unless the solution for $\mu = 0$ is feasible (which can be checked easily), we need to find the value of $\mu$ such that

$$g(\mu) \;=\; \sum_{i=1}^{r} \beta_i \mathbf{1}^T x_i[\mu] - k_s \;=\; 0,$$

that is, we need to find a root of $g(\mu)$ (if $\mu > \tilde{\mu}$ then the $x_i[\mu]$ are all 1-sparse).

In section-3 we explored possible ways to find this root of $g(\mu)$, including using bisection and Newton's method. The reason to use bisection is mostly to avoid points of non-differentiability and local stagnation of Newton's method because of discontinuity. The discontinuous points can be pre-computed and corresponds to the largest entries of the $c_i$'s when they are not uniquely attained. We have denoted $\mathcal{D}$ the set of discontinuous points of $g(\mu)$ in Algorithm 3: if a discontinuous point is encountered, the algorithm returns $\mu$ such that $g(\mu + \epsilon\mu) < 0 < g(\mu - \epsilon\mu)$. Note that the accuracy $\epsilon$ does not need to be high in practice, say 0.001, since there will not be a significant difference between a vector of sparsity $s$ and sparsity $s \pm 0.001$.

Finally, we also provide a fast vectorized GPU-compatible implementation of the Algorithm 1 in PyTorch along with the supplementary materials, where we parallelly project all the vectors together for application in deep neural networks. Since, the Newton's method converges quickly (see Table 3), empirically we observed a fast execution of the the projection of deep networks.

### A.4 Computational Cost

As detailed in Section 3.4, the computational cost of the algorithm is $\mathcal{O}(tN)$, where $N = \sum_{i=1}^{r} n_i$ and $t$ is the number of iterations in Newton's method. The computational cost per iteration is therefore linear in the size of the problem. In practice, we observed that Netwon's method converge very fast and does not require much iterations, as showcased by the following synthetic data example. Let us take $n_i = 1000$ for all $i \in [\![1, 100]\!]$ and generate each entry of the $c_i$'s using the normal distribution $N(0, 1)$. For each sparsity level, we generate 100 such data points and Table 3 reports the average and maximum number of iterations needed by Algorithm 1. In all cases, it requires less than 4 iterations for a target accuracy of $\epsilon = 10^{-4}$.

| | $s_0$ | $s = 0.7$ | $s = 0.8$ | $s = 0.9$ | $s = 0.95$ | $s = 0.99$ |
|---|---|---|---|---|---|---|
| Average | 20.86% | 3.88 | 3.78 | 3.98 | 3.75 | 3.77 |
| Maximum | 21.01% | 4 | 4 | 4 | 4 | 4 |

Table 3: Average and maximum number of iteration for Algorithm 1 to perform grouped sparse projection of 100 randomly generated vectors of length 1000, with precision $\epsilon = 10^{-4}$. The column $s_0$ gives the average and maximum initial sparsity of the 100 randomly generated vectors $x_i$'s.

## B Grouped Sparse Projection

In this section we detail the calculations in the proof of Corollary 3.3.

*Proof.* If $\mu = 0$ and $\mu > \tilde{\mu}$, the projection is unique provided the largest and second entry of each $|c_i|$ are not equal (see Section 3.3). For $0 < \mu < \tilde{\mu}$, since the optimal $\mu^*$ is unique it suffices to prove $x_i[\mu]$ is a one-to-one

mapping, i.e., that if $x_i[\mu_1] = x_i[\mu_2]$ then $\mu_1 = \mu_2$. As noted in Section 3.2, when $|c_i| - \mu\beta_i\mathbf{1} \leq 0$, the solution $x_i[\mu]$ is unique — corresponding to the largest entry of $|c_i|$ — as long as the largest entry is unique.

We now analyse the last scenario, in which $|c_i| - \mu\beta_i\mathbf{1} > 0$ (with $0 < \mu < \tilde{\mu}$). For simplicity of notation, we assume all indexes $j$ of the vector $|c_i|$ are active, i.e., $|c_i(j)| - \mu\beta_i > 0 \ \forall j$. The derivations would follow in an analogous way in the case of some indexes being zero-d out, with sums and norms only including the active indexes. Let's consider $\mu_1, \mu_2$ such that $0 < \mu_1, \mu_2 < \tilde{\mu}$, and consider the $j$-th index of $x_i[\mu_1]$ and $x_i[\mu_2]$ respectively.

$$
\begin{aligned}
&x_i[\mu_1](j) = x_i[\mu_2](j) \\
&\implies \frac{|c_i(j)| - \mu_1\beta_i}{\||c_i| - \mu_1\beta_i\|_2} = \frac{|c_i(j)| - \mu_2\beta_i}{\||c_i| - \mu_2\beta_i\|_2} \\
&\implies (|c_i(j)| - \mu_1\beta_i)^2 - (|c_i| - \mu_2\beta_i)^T(|c_i| - \mu_2\beta_i) = (|c_i(j)| - \mu_2\beta_i)^2(|c_i| - \mu_1\beta_i)^T(|c_i| - \mu_1\beta_1) \\
&\implies [c_i(j)^2 + \mu_1^2\beta_i^2 - 2|c_i(j)|\mu_1\beta_i] \left( \|c_i\|_2^2 + \mu_2^2\beta_i^2 n - 2\mu_2\beta_i|c_i|^T e \right) = \\
&\qquad [c_i(j)^2 + \mu_2^2\beta_i^2 - 2|c_i(j)|\mu_2\beta_i] \left( \|c_i\|_2^2 + \mu_1^2\beta_i^2 n - 2\mu_1\beta_i|c_i|^T e \right) \\
&\implies c_i(j)^2\beta_i^2 n(\mu_2^2 - \mu_1^2) + 2c_i(j)^2\beta_i|c_i|^T e(\mu_1 - \mu_2) + \beta_i^2\|c_i\|_2^2(\mu_1^2 - \mu_2^2) + 2\mu_1\mu_2\beta_i^2|c_i|^T e(\mu_2 - \mu_1) \\
&\qquad + 2|c_i(j)|\|c_i\|_2^2\beta_i(\mu_2 - \mu_1) + 2\mu_1\mu_2|c_i(j)|\beta_i^3 n(\mu_1 - \mu_2) = 0 \\
&\implies (\mu_2^2 - \mu_1^2)\beta_i^2 \left[ c_i(j)^2 n - \|c_i\|_2^2 \right] \\
&\qquad + 2\beta_i(\mu_1 - \mu_2) \left[ c_i(j)^2\|c_i\|_1 - \mu_1\mu_2\beta_i^2\|c_i\|_1 - |c_i(j)|\|c_i\|_2^2 + \mu_1\mu_2|c_i(j)|\beta_i^2 n \right] = 0 \qquad (12) \\
&\implies (\mu_2^2 - \mu_1^2)\beta_i^2 \left[ c_i(j)^2 n - \|c_i\|_2^2 \right] + 2\mu_1\mu_2\beta_i^3(\mu_1 - \mu_2) \left[ |c_i(j)|n - \|c_i\|_1 \right] \qquad (13) \\
&\qquad + 2\beta_i(\mu_1 - \mu_2) \left[ c_i(j)^2\|c_i\|_1 - |c_i(j)|\|c\|_2^2 \right] = 0 \\
&\implies (\mu_1 - \mu_2) \left[ (\mu_1 + \mu_2)\beta_i^2 \underbrace{\left[ \|c_i\|_2^2 - c_i(j)^2 n \right]}_{(A)} + 2\mu_1\mu_2\beta_i^3 \underbrace{\left[ |c_i(j)|n - \|c_i\|_1 \right]}_{(B)} + 2\beta_i \underbrace{\left[ c_i(j)^2\|c_i\|_1 - |c_i(j)|\|c\|_2^2 \right]}_{(C)} \right] = 0.
\end{aligned}
$$

For the equation above to be equal to zero, given $\mu_1, \mu_2, \beta_i > 0$, either the terms $(A) = (B) = (C) = 0$ or $\mu_1 = \mu_2$. Note that if $(A)$ and $(B)$ are equal to 0 then $(C) = 0$, hence we just need to check when $(A) = (B) = 0$. We have that:

$$
\begin{cases} (A) = 0 \\ (B) = 0 \end{cases} \implies \begin{cases} \|c_i\|_2^2 - c_i(j)^2 n = 0 \\ |c_i(j)|n - \|c_i\|_1 = 0 \end{cases} \implies \begin{cases} c_i(j) = \frac{\|c_i\|_2}{\sqrt{n}} \\ |c_i(j)| = \frac{\|c_i\|_1}{n}. \end{cases}
$$

This implies the vector $c_i$ needs to have all entries equal to the same value in every active index $j$. As by assumption this cannot be case — the largest entry of $c_i$ needs to be unique and distinct — we have that $\mu_1 = \mu_2$ and we have proved that $[\mu]$ is a one-to-one mapping.

$\square$

## C   Weighted Grouped Sparse Projection

### C.1   Weighted Sparsity

In section 4 we introduced the notion of *weighted sparsity*: For a vector $x \in \mathbb{R}_0^n$ and given $w \in \mathbb{R}_{0,+}^n$, as following

$$
\mathrm{sp}_w(x) = \frac{\|w\|_2 - \frac{\|Wx\|_1}{\|x\|_2}}{\|w\|_2 - \min_i w(i)} \in [0, 1], \quad \text{where } W = \mathrm{diag}(w)\mathbf{1}. \qquad (14)
$$

Let us make some observations about this quantity

- We have

$$\|w\|_2 = \max_{\|y\|_2 \le 1} \|Wy\|_1 \quad \text{and} \quad \min_i w(i) = \min_{\|y\|_2 = 1} \|Wy\|_1,$$

  which implies the fact that $\text{sp}_w(x) \in [0, 1]$ for any $x$.

- For $w = \mathbf{1}$, we have $\|W(\cdot)\|_1 = \|(\cdot)\|_1$, $\|w\|_2 = \sqrt{n}$ and $\min_i w(i) = 1$ so that $\text{sp}(.) = \text{sp}_{\mathbf{1}}(.)$.

- $\text{sp}_w(x) = 1$ if and only if $x$ is 1-sparse and its non-zero entry corresponds to (one of) the smallest entry of $w$. Therefore, a vector $x$ can be 1-sparse but have a low weighted sparsity $\text{sp}_w(x)$ if the corresponding entry of $w$ is large.

- $\|Wx\|_1$ is a norm if and only if $w > 0$, which we do not require in this paper but use this notation for simplicity.

This notion of weighted sparsity allows to give more or less importance to the entries of $x$ to measure its sparsity. For example, having $w(j) = 0$ means that there is no need for $x(j)$ to be sparse (that is, close or equal to zero) as it is not taken into account in $\text{sp}_w(x)$, while taking $w(j)$ large will enforce $x(j)$ to be (close to) zero if $\text{sp}_w(x)$ is large (that is, close to one). In the remainder of this paper, we will say that a vector $x$ is $w$-sparse if it has a large weighted sparsity $\text{sp}_w(x)$. The following example provides a potential application of projecting a set of vectors onto $w$-sparse vectors.

## C.2 Weighted Group Sparse Projection

We defined the $w$-sparse projection problem of the set of vectors $\{c_i \in \mathbb{R}^{n_i}\}_{i=1}^r$ in (WGSP), where $\{w_i \in \mathbb{R}_{0,+}^{n_i}\}_{i=1}^r$ was the nonzero nonnegative weight vectors associated with the $c_i$'s with a target average weighted sparsity $s_w \in [0, 1]$.

As for 5, the maximum of (WGSP) is attained. Similarly, as for (5), we can derive formula for $x_i$ depending on the Lagrange variable $\mu$ associated with the inequality constraint:

1. For $|c_i| - \mu \beta_i^w w_i \not\le 0$, we have

$$x_i[\mu] = \frac{\left[|c_i| - \mu \beta_i^w w_i\right]_+}{\left\|\left[|c_i| - \mu \beta_i^w w_i\right]_+\right\|_2}.$$

2. For $|c_i| - \mu \beta_i^w w_i \le 0$, $x_i[\mu]$ is 1-sparse. The nonzero entry of $x_i[\mu]$ corresponds to the largest entry of $|c_i| - \mu \beta_i^w w_i$. In this case, since the entries of $w_i$ might be distinct, this non-zero entry does not necessarily correspond to the largest entry of $|c_i|$ and may change as $\mu$ increases. In particular, for $\mu$ large enough, the nonzero entry of $x_i[\mu]$ will correspond to the smallest entry of $w_i$.

We need $\sum_{i=1}^r \beta_i^w w_i^T x_i[\mu]$ to be smaller than $k_s^w$, that is,

$$g_w(\mu) = \sum_{i=1}^r \beta_i^w w_i^T x_i[\mu] - k_s^w \le 0.$$

If $g_w(0) \le 0$, then $x_i[0] = \frac{|c_i|}{\|c_i\|_2}$ is optimal and the problem is solved (the $c_i$'s have average $w$-sparsity larger than $s_w$). Otherwise, the constraint will be active and we need to find a root $\mu^*$ of $g_w(\mu)$. Unfortunately, as opposed to GSP, the function $g_w(\mu)$ is not necessarily strictly decreasing for $\mu$ sufficiently small: when $|c_i| - \mu \beta_i^w w_i \le 0$ and as $\mu$ increases, the $w$-sparsity of $c_i$ may change abruptly as the index of the maximum entry of $|c_i| - \mu \beta_i^w w_i$ may change as $\mu$ increases. In that case, the term corresponding to $c_i$ in $g_w(\mu)$ is piece-wise constant. For example, let $c_i = [4, 1]$ and $w_i = [2, 1]$, we have

- For $\beta_i^w \mu \in [1, 2)$, $\text{sp}_w(x_i[\mu]) = \text{sp}_w([1, 0]) = \frac{\sqrt{5}-2}{\sqrt{5}-1} = 0.19$.

- For $\beta_i^w \mu \ge 2$, $\text{sp}_w(x_i[\mu]) = \text{sp}_w([0, 1]) = 1$.

However, when $|c_i| - \mu \beta_i^w w_i \not\le 0$ for some $i$, $g_w(\mu)$ is strictly decreasing.

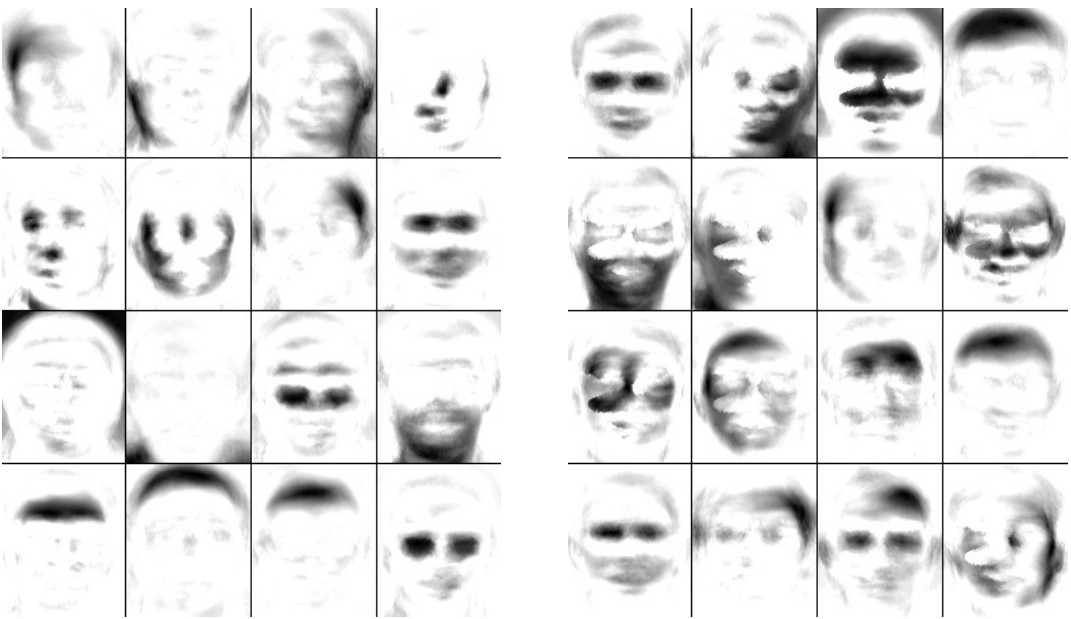

Figure 3: Basis elements obtained with sparse NMF (left) and weighted sparse NMF (right).

### C.2.1   Example Application on Facial Images

Let us consider the case when the $x_i$'s represent a set of basis vectors for (vectorized) facial images. In this case, one may want these basis vectors to be sparser on the edges since edges are less likely to contain facial features; see, e.g., Ho (2008). Let us illustrate this on the ORL face data set (400 facial images, each 112 by 92 pixels). Each column of the input data matrix Y represents a vectorized facial image, and is approximated by the product of XH where X and H are nonnegative. Each column of $W$ is a basis elements for the facial images. We first apply sparse NMF with average sparsity of 60% for the columns of basis matrix X. The basis elements are displayed on the left of Figure 3.

Given a basis image 112 by 92 pixels, we define the weight of the pixel at position $(i, j)$ as $e^{\|(i,j)-(56.5,46.5)\|_2/\sigma}$ with $\sigma = 5$ as in Chapter 6 of Ho (2008). The further away from the middle of the image, the more weights we assign to the pixel so that the basis images are expected to be sparser on the edges. According to these weights, the average weighted sparsity of the sparse NMF solution is 89% (in fact, we notice that most basis images are already relatively sparse on the edges). Then, we run weighted sparse NMF (WSNMF) with average weighted sparsity 95%. The basis elements are displayed on the right of Figure 3. We observe that, as expected, the edges are in average even sparser compared to the unweighted case (note that the only WSNMF basis element that is not sparse on the edges, the third image, has darker pixels in the middle of the image to compensate for the relatively lighter pixels on the edges). This is confirmed by Figure 4 which displays the average of the columns of the squared error for both solutions, that is, it displays the average of the columns of the squared residual $(Y - XH)^2_{ij}$. The residual of WSNMF is brighter on the middle of the image (since the basis elements are less constrained to be sparse in this area) and darker at the corners. In fact, most of the error of WSNMF is concentrated in the four corners.

Observe also that the basis elements of WSNMF are denser: in fact, the (unweighted) sparsity of WSNMF is only 50%. The relative errors, that is, $\frac{\|Y-XH\|_F}{\|Y\|_F}$, for both approaches are similar, 20.34% for sparse NMF and 20.77% for WSNMF.

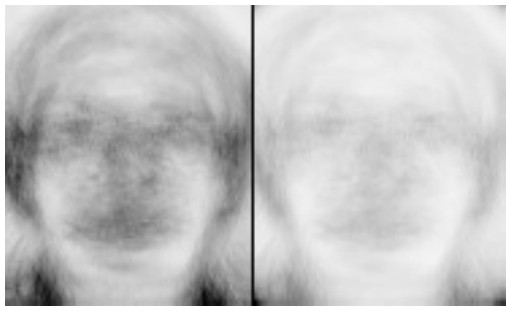

Figure 4: Average squared error obtained with sparse NMF (left) and WSNMF (right) –the darker, the higher the error.

### C.3 Theoretical Results

We now provide theoretical proofs for the solutions of the problem 4, as well as proof of Theorem 3.2 for GSP. Lemma C.1 provides a proof for the solution of the dual problem for a general weighted sum of $x[\mu]$, from which both results follow.

**Lemma C.1.** *Let $w \in \mathbb{R}_{0,+}^n$ and $x \in \mathbb{R}_+^n$. Let also $f_x(\gamma) = w^T x[\gamma]$ where*

- *If $x - \gamma w \nleq 0$, that is, if $\gamma < \tilde{\gamma} = \max_j \frac{x(j)}{w(j)}$, then*

$$\bar{x}[\gamma] = \frac{[x - \gamma w]_+}{\|[x - \gamma w]_+\|_2}.$$

- *Otherwise $x[\gamma]$ is a 1-sparse vector, with its nonzero entry equal to one and at position $j \in \arg\max_j x(j) - \gamma w(j)$.*

*For $0 \le \gamma < \tilde{\gamma}$, $f_x(\gamma)$ is strictly decreasing, unless $x$ is a multiple of $w$ in which case it is constant. For $\gamma \ge \tilde{\gamma}$, $f_x(\gamma)$ is nonincreasing and piece-wise constant.*

*Proof.* The case $\gamma \ge \tilde{\gamma}$ is straightforward since $f_x(\gamma) = w_j$ for $j \in \arg\max_j (x(j) - \gamma w(j))$: as $\gamma$ increases, the selected $w_j$ can only decrease since $w \ge 0$.

Let us now consider the case $0 \le \gamma < \tilde{\gamma}$. Clearly, $f_x(\gamma)$ is continuous since it is a linear function of $x[\gamma]$ which is continuous, and it is differentiable everywhere except for $\gamma = \frac{x(j)}{w(j)}$ for some $j$. Therefore, it suffices to show that $f'(\gamma)$ is negative for all $\gamma \ne \frac{x(j)}{w(j)}$. Note that $f(\gamma)$ is strictly decreasing if and only if $c_1 f(\gamma c_2)$ is strictly decreasing for any constants $c_1, c_2 > 0$. Therefore, we may assume without loss of generality (w.l.o.g.) that $\|w\|_2 = 1$ (replacing $w$ by $w/\|w\|_2$). We may also assume w.l.o.g. that $x > \gamma w$ otherwise we restrict the problem to $x(J)$ where $J(\gamma) = \{j | x(j) - \gamma w(j) > 0\}$ since $f$ depends only on the indices in $J$ in the case $\gamma < \tilde{\gamma}$. Under these assumptions, we have

$$f(\gamma) = \frac{w^T(x - \gamma w)}{\|x - \gamma w\|_2} = \frac{w^T x - \gamma}{\|x - \gamma w\|_2},$$

since $\|w\|_2^2 = w^T w = 1$, and

$$f'(\gamma) = \frac{-\|x - \gamma w\|_2 + (w^T x - \gamma)\, w^T(x - \gamma w)\|x - \gamma w\|_2^{-1}}{\|x - \gamma w\|_2^2}$$

$$= \frac{-\|x - \gamma w\|_2^2 + (w^T x - \gamma)^2}{\|x - \gamma w\|_2^3}.$$

It remains to show that $||x - \gamma w||_2^2 \geq (w^T x - \gamma)^2$ implying $f'(\gamma) < 0$. We have

$$||x - \gamma w||_2^2 = ||x||_2^2 - 2\gamma w^T x + \gamma^2,$$

and

$$(w^T x - \gamma)^2 = (w^T x)^2 - 2\gamma w^T x + \gamma^2,$$

which gives the result since $|w^T x| < ||w||_2 ||x||_2 = ||x||_2$ for $x$ not a multiple of $w$. $\qquad\square$

*Proof of Theorem 1.* The proof follows from Lemma C.1 by setting $w = \mathbf{1}$ and noting that $g(\mu) = \sum_{i=1}^{r} f_{|c_i|}(\beta_i \mu) - k_s$. Note that since $w(j) = 1 \; \forall j$, the index of the maximum element remains the same. $\quad\square$

**Corollary C.2.** *The function $g_w(\mu) = \sum_{i=1}^{r} \beta_i^w w_i^T x_i[\mu] - k_s^w$ as defined above is nonincreasing. Moreover, if $|c_i| - \mu \beta_i^w w_i \nleq 0$ for some $i$, it is strictly decreasing.*

*Proof.* This follows from Lemma C.1 since $g_w(\mu) = \sum_{i=1}^{r} f_{|c_i|}(\beta_i^w \mu) - k_s^w$. $\qquad\square$

Therefore, as opposed to $g(\mu)$, $g_w(\mu)$ could have an infinite number of roots $\mu^*$. However, the corresponding $x_i(\mu^*)$ is unique so that non-uniqueness of $\mu^*$ is irrelevant. Moreover, this situation is rather unlikely to happen in practice since it requires $|c_i| - \mu \beta_i^w w_i \leq 0$ for all $i$ at the root of $g_w(\mu)$ hence it requires $s_w$ to be close to one (that is, $k_s^w$ to be large). Similarly as for $g$, $g_w$ will be discontinuous at the points where $\max_j \frac{c(j)}{\beta_i w_i(j)}$ is not uniquely attained: as $\mu$ increases, the two (or more) last non-zero entries of $x_i[\mu]$ become zero simultaneously. Finally, to solve WGSP, we can essentially use the same algorithm as for GSP, that is, we can easily adapt Algorithm 1 to find a root of $g_w(\mu)$.

# D   Sparse NMF: Experiment Details

In this section, we provide experiment parameters and details for the sparse NMF experiments.

## D.1   Synthetic data sets

For the experiment on synthetic data, we take $m = n = 100$ and $r = 10$. We generate each entry of $X$ using the normal distribution with mean 0 and variance 1 and then set the negative entries to zero, so that $X$ will have sparsity around 50%. We generate each entry of $H$ using the normal distribution in the interval $[0, 1]$. To run gspNMF and cgspNMF, we compute the average sparsity of the columns of the true $X$ and use it as an input. We generate 50 such matrices, and use the same initial matrices for all algorithms, which were generated using the uniform distribution for each entry. In Figure 1 of the paper, we reported the evolution of the average of the relative error obtained by the different algorithms among the 50 randomly generated matrices.

## D.2   Image data set

For this set of experiments we test on the widely used data set in the NMF literature, the CBCL facial images. This dataset consists 2429 images of $19 \times 19$ pixels, and was used in the seminal paper by Lee & Seung (1999) with $r = 49$. We run the sparse NMF techniques with sparsity 85% Using 10 random initializations, Figure 1 (in paper) reports the evolution of the average relative error, and Figure 5 displays the basis elements obtained by different methods.

# E   Deep Network Pruning with GSP: Experiments, settings and hyperparameters

## E.1   Experiments on CIFAR-10

We use the CIFAR-10 dataset (Krizhevsky et al., 2009) to train and test the VGG16, Resnet-56 and Resnet-110 models for our experiments. The CIFAR-10 dataset was accessed through the dataset API of the torchvision

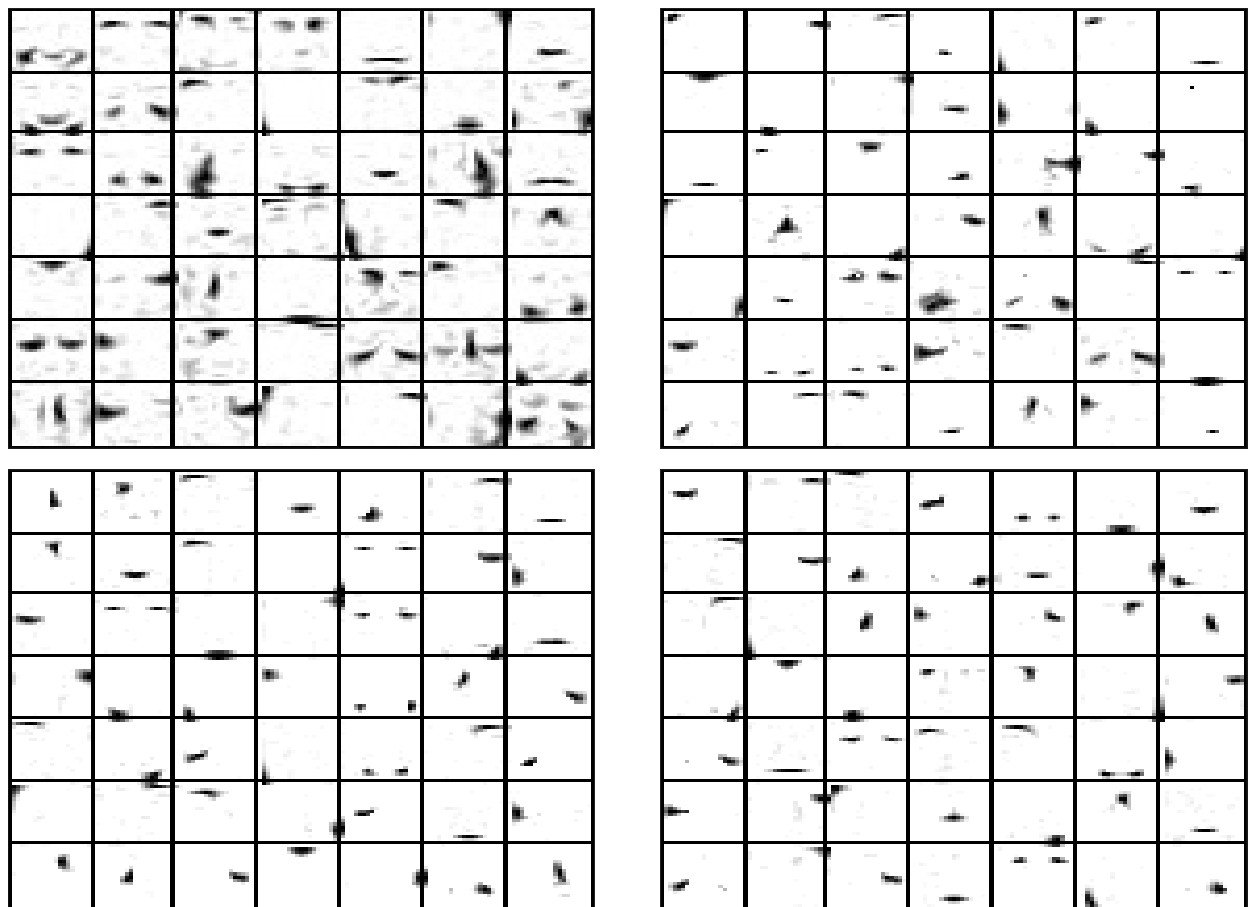

Figure 5: Basis elements obtained with NeNMF (top left), PSNMF with sparsity 0.85 (top right), $\ell_1$ A-HALS with sparsity 0.85 (bottom left), cPSNF with sparsity 0.85 (bottom right).

package of PyTorch. We performed the standard preprocessing on the data which included horizontal flip, random crop and normalization on the training set. With the CIFAR-10 dataset we perform two different types of experiments. First, we perform an experiment with layerwise induced GSP integrated with the training phase. We also perform single shot pruning with a single projection of the model weights followed by the finetuning phase in section 5.3.

For the experiments with intermittent projections during the training phase, we project the weights of the VGG16 model using GSP with sparsity level $s$, perform a forward pass on the projected weights and finally update the model parameters using backpropagation every 150 iterations for 200 epochs, starting from epoch 40. We reduce the learning rate by a factor of 0.1 at the milestone epochs of 80, 120 and 160. Next, we set the $s$ fraction of the lowest parameters of the model to zero. At this point the model is sparse with a layerwise *hoyer sparsity* of $s$. However, since we project intermittently and with *hoyer − sparsity* being a differentiable approximation to the $\ell_0$ norm, we then prune the surviving weights that are close to zero or zero, keeping the largest $1 - \tilde{s}$ fraction of the parameters, where $\tilde{s}$ is the final sparsity of the model. We fix these pruned parameters (do not train them) with a mask. Finally, we finetune the surviving parameters for 200 epochs with a learning rate of 0.01 and dropping the rate by 0.1 in the same milestones as the sparsity inducing run. In the case of single shot GSP, we take a pretrained model, make a single projection of the layers with a sparsity $s$ and then prune and finetune the model with similar parameters as above.

### E.2 Experiments on ImageNet

In these set of experiments, we use the ImageNet dataset (Russakovsky et al., 2015) to train and test the Resnet-50 model for our experiments. ImageNet is augmented by normalizing per channel, selecting a patch with a random aspect ratio between 3/4 and 4/3 and a random scale between 8% to 100%, cropping to 224x224, and randomly flipping horizontally.

For the experiments with induced-GSP, we project the weights of the ResNet50 model in a similar technique to the experiments performed with the CIFAR-10 dataset. We first project the layers of the model using GSP with $s = 0.80$, perform a forward pass on the projected weights and finally update the model parameters using backpropagation every 500 iterations for 120 epochs. We start the projection of the model from epoch 40 and keep projecting every 500 iterations till the final epoch. In both the cases of CIFAR-10 and ImageNet we choose the iteration interval of projection in such a way so that there are 3 projections per epoch. We reduce the learning rate by a factor of 0.1 at the milestone epochs of 70 and 100. Next, we set the $s$ fraction of the lowest parameters of the model to zero. We next prune the surviving weights that are close to zero or zero, keeping the largest $1 - \tilde{s}$ fraction of the parameters, where $\tilde{s}$ is the final sparsity of the model. Finally, we finetune the surviving parameters for 140 epochs with a learning rate of 0.001 and dropping the rate by 0.1 in the same milestones as the inducing-GSP run.

