# OpenReview forum: "Explicit Group Sparse Projection with Applications to Deep Learning and NMF"
_TMLR — Accepted by TMLR_

### Review · Reviewer_QiA9 · 2022-08-04

**Summary Of Contributions:**

This manuscript proposes a group sparse projection algorithm based on the Hoyer index, which relates to the sparsity of a vector. In particular, the projection is defined with a constraint set that measures the average Hoyer index of a set of vectors. The manuscript proposes an optimization algorithm that optimizes the dual formulation, scaling linearly in the problem dimension. The authors compare their approach for sparse non negative matrix factorization as well as pruning of artificial neural networks' weights.

**Broader Impact Concerns:**

NA.

**Requested Changes:**

**Main comments**

**[Notation]**
Respectfully, the paper does not look in a condition for review, as there are multiple inconsistencies in notation. For example, in the paragraph titled Notation the authors mention that they use $x(j)$ to denote the j-th entry in the vector $x$. *On the following sentence* they refer to the i-th entry in $x$ as $x_i$. These issues persist (and increase) throughout the paper: e.g. in Sect 3.3, point (a), the authors write $x_i(0)$ to denote *a vector*, as they seem to be referring to the i-th of $r$ vectors as a function of the scalar (Lagrange parameter) $\mu$. It is very hard to follow the paper and understand what is being done with all these inconsistencies. Please, use conventional notation (using $e$ to denote a vector of all ones is *very confusing*).

**[Definition of sparsity]**
The authors seem to argue (throughout the introduction) that their choice of sparsity measures (the Hoyer index) is more "interpretable" and that it does not need a parameter search to achieve a desired sparsity level. This is incorrect: a Hoyer measure of 0.5 does not imply that 50% of the entries will be zero, and if one desires 50% of zero entries one indeed needs to carry out a parameter search over the Hoyer measure. It is true that the Hoyer measure, sp(x), is nice in that sp(x) implies that the vector is fully dense, and sp(x)=1 implies that x only has 1 non-zero element. However, a similar conclusion can be drawn by using an $\ell_1$ norm: if one computes the proximal of the $\ell_1$ norm at a point $x$ with a penalty parameter of $\lambda \in [0,\lambda_{max}]$, where $\lambda_{max}=\lVert x \rVert_{\infty}$ , then $\lambda=0$ leads to no sparsity enforced, and $\lambda = \lambda_{max}$ leads to only 1 non-zero (maximal sparsity). In light of this, the "benefits" over and $\ell_1$ to measure sparsity are unclear (and exaggerated). Please, do let me know if there's another benefit that I'm failing to see.

**[Organization]**
The organization of the paper is unclear, and many things are undefined:
* The introduction section is very short, and it would be beneficial for the reader to have an introduction to the use of sparsity in inverse problems and compression, as well as some background on the Hoyer measure.
* The paragraph on *Projections* is very strange: "projections" is a mathematical operator that is much broader than what the authors mention in their paragraph. Perhaps the authors are rather referring to the use of regularization (either as projections or regularization) in inverse problems. I would include this in the Introduction.
* The authors then present "Sparse NMF". Yet, their definition of the problem in Eq. (1) is not a sparse NMF problem, as there is no sparsity enforced at all. It is true that the authors mention at the end of that paragraph that "In practice, it is particularly useful to have sparse X and/or H to have a more interpretable decomposition". *(i)* enforcing sparsity is not just an ad-hoc practical trick, but has been largely studied theoretically. The benefit sparse components is not just "interpretability" but rather leading to statistical benefits. *(ii)* the authors never formalize what the sparse NMF problem is, as they never write any sparsely-regularized version of the problem in Eq. (1). This becomes problematic later in section 5.1 when the authors write "As H is not affected by the sparsity requirement..." : what sparsity requirement? up to this point, there's no sparse-enforcing term on either factor.

**[Experimental results]**
* The experimental section is unclear. Admittedly, part of this is likely because the presentation of the method is not fully clear (for the issues above), but also because the authors present and compare many methods that have not been formally defined, and so one does not know what these comparisons really mean. Importantly, some of these methods penalize $\ell_1$ norms, some penalize Hoyer measures, and it's not very clear which ones do what. It'd be very useful to *define* the problems that each method solve (indeed, these were never defined).
* When comparing methods that enforce sparsity in different ways ($\ell_1$ and $sp$), is it not clear how the choices for these regularizers were done (i.e, what $\lambda$, or what value of $s$), or how they were chosen. Were they optimized for each method independently? Were they set so that they all achieve the same level of effective sparsity (i.e. cardinality)? Without this information, it is hard to understand the meaning of these graphs (e.g. the lines that converge to an error of >0.1 might have parameters that enforce too much sparsity, thus resulting in higher errors).
* The authors mention compare all algorithms in terms of error per iteration, "Since all algorithms have almost the same computational complexity per iteration", yet, the computational cost per iteration is not the same for the different methods (as per footnote 1). Thus, a comparison of runtime would also be appropriate here.
* It is a bit strange that the sparsity percentages (on Table 2) have no decimal points.

**[other main comments]**
* The authors mention that the maximum of Eq. (5) is attained since the objective function is continuous and the feasible set compact. I am unsure this is true, since the problem seems to optimize over vectors different from the zero vector (indeed, see the problem (GSP)). Thus, $\mathbb R^n_0$ is not compact.

**[Typos and other errors]**
* There are many typos throughout the manuscript. E.g., in the *Notations*, the paper have two different summations over the variable $i$, but use $j$ to index the summands.
* In many equations (e.g. in GSP, eq 5, eq 6, among many other places) the authors write a maximum operator over the vectors $x_i$ as well as the indices $i$, i.e.: $\max_{x_i,1\leq i\leq r}$. However, the argument of this operator has no index on $i$, because there is a summation from $i=1$ to $i=r$.
* At the top of page 5, when "reformulating" their GSP problem, the authors are missing the $\ell_2$ norms in their expressions for $sp(x_i)$.
* The authors comment (Sec 2.1) that the Hoyer index is smooth. This statement is incorrect: first, this index is discontinuous (at x=0). Second, the $\ell_1$ norm (and thus, $sp(x)$) is not smooth at points wherever any of the entries in $x$ are zero (even if the overall vector is non-zero).
*  Consider including the constraint of $n>1$ in the definition in Sec. 2.1.
* Denoting by $\lVert x \rVert_w$ the weighted $\ell_1$ norm of $x$ is very confusing, as this is typically the notation used for weighted $\ell_2$ norms. Consider using $\lVert W x \rVert_1$, where $W = \text{diag}(w)$.


**Strengths And Weaknesses:**

**Strenghts**
- Optimizing an average notion of "sparsity" based on the Hoyer index is interesting.
- Numerical results seem to suggest benefits of their approach.

**Weaknesses**
- The manuscript is unclear, with contradicting notation throughout the paper.
- Comparing methods are unclear (or in some cases not defined at all), making the interpretation of the results difficult.
- Some statements are hand-wavy or exaggerated.

---

> ### Author Response · Authors · 2022-09-05
> **Response to Reviewer QiA9**
>
> We sincerely thank the reviewer for the careful and detailed review. We appreciate that the reviewer found interesting the approach of optimizing an average notion of sparsity based on the Hoyer index. Below, we address the changes requested by the reviewer and provide our responses. We have also updated the manuscript to reflect the changes the reviewer required (all changes are indicated in blue color).
>
> **Notation**
> We thank the reviewer for pointing out inconsistencies in our notation. We have now carefully checked and corrected our notation; in particular, given a vector $x$, $x(i)$ denotes its $i$th entry, $\bf{1}$ is the vector of all ones, $x[\mu]$ is the solution of our problem depending on the parameter $\mu$.
>
> *Remark*: The notation $e$ for the vector of all ones is a standard notation in the numerical linear algebra community; see, e.g., [http://ee263.stanford.edu/notes/matrix-primer-lect1.pdf](https://), or [https://www.jstor.org/stable/pdf/41062018.pdf](https://)
>
> **Definition of Sparsity**
> We agree with the reviewer. We do not claim that the measure itself is more interpretable, but that it has many nice properties, as discussed at length for example in the papers of Hurley \& Rickard (2009) and Thom et al. (2015); see also Appendix A.1. In particular, Hoyer sparsity measure is continuous except at zero, e.g., the vector $[1, 10^{−6}, 10^{−6}]$ is sparser than $[1, 1, 0]$ which is meaningful numerically. We have clarified this in the updated manuscript.
>
> **Organization**
> 1. We agree the Introduction was rather short. We have tried our best to make it clearer, and in particular added the definition of the Hoyer measure and explaining better what our paper is aiming to do.
> 2. We agree, and have renamed that paragraph.
> 3. We agree with the reviewer that our paper was lacking some details and had some confusing statements.
> 	We have now significantly improved the parts about sparse NMF (both in Section 1.1 and Section 5.1); in particular providing the explicit formulations we are working with; see Section 5.1.
>
> **Experimental Results**
> 1. As mentioned above, we have done our best to clarify this part.
> 2. We agree it is not easy to compare methods based on different sparsity inducing approaches.
>     To summarize, for sparse NMF we compare to $\ell_1$ A-HALS which penalized the $\ell_1$ norm; see the new formulation (10) from the paper (Gillis, JMLR, 2012). We use the code made available: the penalty parameters are tuned automatically to achieve a desired $\ell_0$ sparsity level for each column of $\mathrm{X}$. In spirit, this method is similar to the cspNMF that projects each column of $\mathrm{X}$ independently to have a given Hoyer sparsity. We have clarified this in the paper. We have also added a footnote discussing the differences in these sparsity measures. It turns out the Hoyer measure is often close to the $\ell_0$ measure.
> 3. We have performed such an experiment, and in fact the run times are very close to one another, given that $n$ is sufficiently large. The projection operator is only applied on the $m$-by-$r$ matrix X and runs in linear time. Therefore, it is negligible to the $O(mnr)$ complexity of first-order methods for NMF as long as $n$ is sufficiently large, which we have verified numerically. We have added some clarification in the footnote.
> 4. The sparsity percentages were rounded to match with the other table where no decimal points were used. In the updated manuscript we have now added back the decimal points.
>
>
> **Other Main Comments**
> The feasible set is { $\{ x \in \mathbb{R}^{n} | \,||x||_2 = 1, x \geq 0 \, \text{and} \, sp(x) \geq s \}$ } which is compact and does not contain the 0 vector since $||x||_2 = 1$  (we have removed the $_0$ in $\mathbb{R}^{n}_0$ from the formulation, it was unnecessary and confusing in fact).
>
> **Typos and other errors**
> 1. We have done our best to remove such typos from the paper.
> 2. We used a standard notation in optimization, namely $\max_{x \in \mathcal{X}} f(x)$ such that $x$ satisfies some other constraints, where $\mathcal{X}$ is the set that has to contain $x$, and $f$ is the objective function.
>     However, we have clarified by introducing the set $\boldsymbol{\mathcal{X}} = \{ x_i \in         \mathbb{R}_0^{n_i}, i \in [[1,r ]] \, : \, x_i \geq 0, ||x_i||_2 = 1 \, \forall i \}$
>     over which we optimize over.
> 3. Since we optimize over $\boldsymbol{\mathcal{X}}$ which requires $||x_i||_2 = 1$, the $\ell_2$ norm can be dropped. We have clarified in the paper.
> 4. We agree, and have corrected this error. Note that, on the non-negative orthant, $\text{sp}(x)$ is continuous everywhere except at $x=0$.
> 5. We have included this.
> 6. We have included this.

---

> > ### Comment · Reviewer_QiA9 · 2022-09-19
> > **Manuscript in better shape after revisions**
> >
> > I thank the authors for carefully revising the manuscript based on my comments and that of the other reviewers. I have gone through the other reviewers comments, the corresponding answers and modifications in the paper. I think the manuscript is in significantly better shape now, and I would support its acceptance in TMLR.
> >
> > I have some final small comments in the new added parts:
> >
> > - page 1: "that are close to" -> "that are closest to"
> > - Sec 1.2, "with an single sparsity" -> "with a single sparsity"
> > - Sec 2.1, the authors write that "$sp(x)$ is smooth everywhere except at the origin". I'm not sure this is true - isn't $sp(x)$ non-smooth whenever $x(i)=0$ for any i (while $x\neq 0$)?
> > - Sec 5.1. (second point) "nonnelative" -> "nonnegative"

---

> > > ### Author Response · Authors · 2022-10-15
> > > **Response to Reviewer QiA9**
> > >
> > > We appreciate the reviewer's careful feedback which has greatly helped us to improve the paper during the review process. We have corrected the typos identified by the reviewer. Furthermore, we address a comment by the reviewer below:
> > >
> > > "Sec 2.1, the authors write that sp(x) is smooth everywhere except at the origin. I’m not sure this is true - isn’t sp(x) non-smooth whenever x(i) = 0 for any i (while x ̸ = 0)?"
> > >
> > > >It is not smooth as soon as there is a zero in fact. However, with non-negativity constraints, only the origin is concerned (since $||x||_1 = 1^Tx$ for $x ≥ 0$). Note that it is continuous everywhere except at the origin. We have added a further comment regarding this in the paper.

---

### Review · Reviewer_AMJx · 2022-08-04

**Summary Of Contributions:**

This paper provides an explicit group sparsity projection method based on the Hoyer sparsity measure. Specifically, the paper makes the following contributions:

1. This paper provides a clever formulation of the group sparse projection problem with explicit sparsity control with the Hoyer measure, making the problem faster to solve
2. This paper theortically derives an efficienct algorithm to solve the proposed group sparse projection problem
3. This paper provides emperical justification showing the proposed method is effective on both NMF and neural network pruning applications

**Broader Impact Concerns:**

I don't see any potential concerns for this submission.

**Requested Changes:**

Please refer to the two weakness points mentioned previously. Point 1 on the novelty is a critical issue needs to be addressed in the revision, while point 2 on the comparison would further strengthen the work.

Typo: Section 2.2 line before Equ. (3) "can formulated" -> "can be formulated"

**Strengths And Weaknesses:**

### Strength
1. The paper brings up important research question on performing explicit sparse projection with group sparsity constraint, and provides an effective solution
2. The derivation on the GSP problem formulation and solution is solid
3. The paper shows convincing experimental results on the effectiveness of the proposed method
4. The paper is well written and easy to follow

### Weakness
1. The theoritical contribution needs further clearification. From the description of Section 3.1 and 3.2 seems like the formulation and solution of GSP borrowed technique from the cited (Thom et al., 2015) paper. As this theortical derivation serves the main contribution of this work, it is suggested to clearly identify which part of the formulation or solution is derived from (Thom et al., 2015), and what is unique for the proposed method. Something worth discussing would be (1) if the technique of converting Hoyer to L1 by normalizing the L2 norm in problem formulation is novelly proposed, and (2) what is the major difference in solving for group projection vs. single vector projection?
2. This work mainly focus on using Hoyer measure for sparsity constraint. Meanwhile it would also be possible to use L1 (group LASSO) or L0 (magnitude pruning) as explicit sparsity constraint for the GSP problem, which will lead to simpler and more efficient solutions. It's suggested to explicitly discuss and compare with these schemes.

---

> ### Author Response · Authors · 2022-09-05
> **Response to Reviewer AMJx**
>
> We thank the reviewer for the careful review. We appreciate that the reviewer found the research questions tackled by our manuscript to be important and are also encouraged that the reviewer found the derivations on the GSP problem formulation and solution to be solid and the experimental results to be convincing. We are also happy that the reviewer found the manuscript to be well written and easy to follow.
>
> We address the points raised by the reviewer in the weakness section in the following. We have also updated the manuscript to reflect the changes the reviewer required (all changes are indicated in blue color)
>
> 1. *"The theoritical contribution needs further clearification ..."*
> 	(a) As far as we know, our approach of converting Hoyer to $L_1$ by normalizing the $L_2$ norm in our problem formulation is in fact novel. We have clarified this in the paper.
>
> 	(b) Although the idea behind our derivations are similar, namely using duality (which is a standard approach), the technical details are rather different since we have a single dual variables since the different vectors $x_i$'s depend on each other (there is only one sparsity constraints). In particular, we have to carefully treat the case when some of these vectors are projected onto 1-sparse vectors. We have also clarified this in the paper.
>
>
> 2. *This work mainly focus on using Hoyer measure for sparsity constraint ...*
> 	We agree that there are many other approaches to achieve sparse solutions. However, the focus of this paper is on the Hoyer measure which has been shown to be powerful in many scenarios; see Thom et al. (2015) and the references therein. We believe it is out of scope of this paper to compare all the possible sparsity measures. In the numerical experiments, we compare our approach with one using $L_1$ penalty in the case of sparse NMF, and to several recent state-of-the-art pruning approaches in the case of deep neural networks.
>
> We have also addressed the comments made in the requested changes section by the reviewer in the updated version and have corrected the typo.

---

### Review · Reviewer_Ycco · 2022-08-15

**Summary Of Contributions:**

This paper develops a sparse projection method called Grouped Sparse Projection (GSP) by which a set of feature vectors can be learned to be sparse. The problem is formulated as a sparsity inducing optimization based on the Hoyer sparsity measure. The approach develops further into a dual problem such that the sparsity parameter can be set for a group of vectors without having to learn individually for each vector, in which the authors use Newton’s method to compute $g(\mu)$ and $g'(\mu)$. The authors also provide some variants to fit some different scenarios such as weighted GSP or single-shot GSP. The method is evaluated on nonnegative matrix factorization and neural network pruning tasks and achieves competitive results to the alternative approaches.

**Broader Impact Concerns:**

No concerns I have.

**Requested Changes:**

* Section 1 makes a point about the contribution in a high-level view, but misses some technical details about the proposed methodology.
“Projections” in Subsection 1.1 appears to be incomplete, sounds arbitrary, or doesn’t really deliver much insight into how it is related to the proposed work.
“Contributions” in Subsection 1.2 needs to discuss how Newton’s method is being used.
Also include that the problem is being solved in the dual, and how this is related to the sparsity parameter.
Discuss how significant it is to have the characterization of the optimal solution (that is on the uniqueness).
The proposed method “group sparse projection” is quite vague as it can refer to any existing prior work on group sparsity.

* Section 2 (Background on sparse projection) is a good place to discuss background material, but has a lot of room for improvement.
Subsection 2.1 doesn’t quite work as “problem definition”; instead it explains what is the Hoyer measure and its properties (continuing in the appendix); rather it is defined in Subsection 2.1 where the authors explain the sparse projection as an optimization problem; this section needs organizing.
This section needs to provide more explanations on how the soft thresholding operator has to do with the sparse projection and the Hoyer measure to fill the gap in the background materials.
Perhaps move “Notation” here from the introduction.

* Section 3
Discuss why the problem has to be re-formulated and solved in the dual, perhaps with the link to the sparsity parameter.
Discuss why Newton’s method is necessary and the background on Newton’s method including the initialization, bisection method, etc.
Discuss what it means to have nonzero x (as in the case for neural networks; as opposed to the nonnegative matrix factorization) for the proposed method.
Perhaps also discuss how it compares to the sparse group Lasso (this can be put in the previous sections).

* Section 5
Provide a lot more details on the baselines of NMF. It is not clear whether or not the speedup gain in the convergence comes from the proposed idea of group sparsity or Newton’s method; when it is compared to NeNMF.

* Other technical details
Discuss in detail how one sets the desired sparsity level both in the algorithm and for experiments.

-- After authors' response
I have read the authors response and the update made on the manuscript. My major concern was on clarifying the methodologies, and I think the authors have made quite significant (and positive) changes as to improving the paper.

**Strengths And Weaknesses:**

The paper addresses the sparse learning problem which is important for efficient large-scale and also interpretable machine learning. The paper discusses relevant prior works in the literature adequately. The writing is fine (up to a point before the main methodology), although there are parts missing details. It is interesting that the method can avoid the cost of searching for the optimal regularization parameter and tunes the level of sparsities over a set of vectors quite automatically, although the authors don’t seem to analyze more about how it helps in terms of interpretable feature learning. The single-shot GSP is interesting provided that it does not require the sparsity-inducing optimization step and this can be quite significant from the compression point of view.

---

> ### Author Response · Authors · 2022-09-05
> **Response to Reviewer Ycco**
>
> We sincerely thank the reviewer for the careful review and helpful suggestions for the paper.  We appreciate that the reviewer agrees with the importance of the addressed sparse learning problem for efficient large-scale training. We are also encouraged by the reviewer's emphasis on the potential of the single-shot GSP. Below we address each of the points brought up by the reviewer in the *requested changes* section. We have also updated the manuscript to reflect the changes the reviewer required (all changes are indicated in blue color).
>
> 1. a)*" Section 1 makes a point about the contribution in a high-level view, but misses some technical details ..."*
>
>     We have revised the introduction (Section 1) significantly, as this issue was also raised by another reviewer, as well as address the "Projections" subsection.
>
> 	b) *"Contributions” in Subsection 1.2 needs to discuss how Newton’s method is being used."*
>
>     We have added a new paragraph explaining the use of Newton's method in more detail; see Section 3.4. However, since the Newton's method (combined with bisection for non-smooth functions) is a standard textbook approach to compute the root of a function, namely find $\mu$ such that $g(\mu) = 0$, we have not included this as a main contribution.
>
> 	c) *"Also include that the problem is being solved in the dual ..."*
>
>     We have added a short sentence in the introduction regarding our approach to compute our projection.
>
> 	d) *"The proposed method “group sparse projection” is quite vague as it can refer to any existing prior work on group sparsity."*
>
>     We agree that "grouped sparse projection” can be vague if taken outside of its context. However, in the context of our paper, we believe it is sufficiently clear since we only use it to refer to our approach. If the reviewer insists, we could rename our method, e.g., to "Hoyer grouped sparse projection''.
>
> 2. We agree and have tried our best to better organize these sections. We have for example renamed Section 2.1 as "Hoyer sparsity measure'', we have introduced the Hoyer measure earlier, we have updated the Notation, and reorganized Section 1.
>
> 3. *"Discuss why the problem has to be re-formulated and solved in the dual, ..."*
>
>     As mentioned above, we have added more details and discussions about Newton's method.
>
>     *"Discuss what it means to have nonzero x (as in the case for neural networks; as opposed to the nonnegative matrix factorization) for the proposed method."*
>
>     Having a zero vector, $X(:,i) = 0$ for some $i$ in NMF, means that the factorization rank has been reduced since one basis vector is identically zero. This is typically not a desired result in practice. Moreover, in Deep Neural Networks, having a layer with all zeros lead to what is known as *"layer collapse"*, i.e., it creates a fully pruned layer with all zero parameters, making the network untrainable. We discuss this in Appendix A.2, but we can bring this within the main paper if the reviewer insists.
>
> 4. *"It is not clear whether or not the speedup gain in the convergence comes from the proposed idea of group sparsity or Newton’s method; ..."*
>
>      The main speed up comes from using GSP instead of projection onto the non negative orthant. We are using the NMF algorithm (namely, NeNMF that relies on Nesterov fast gradient method), replacing the projection onto the nonnegative orthant by our grouped sparse projection. Note, the speedup does _not_ come from Newton's method, since Newton's method is only used in the projection step (not directly to solve the NMF problem).
>
> 5. *Other technical details Discuss in detail how one sets the desired sparsity level ...*
>
> 	This is a difficult question, as this highly depends on the application at hand, and the goals of the user. For example, when pruning sparse neural networks, there is a tradeoff between precision and sparsity, which should be carefully assessed by the user; see, e.g., Table 1. In NMF, there is a tradeoff between sparsity and the approximation error; please see Figure 1 (right). We have added a paragraph discussing this issue; see the conclusion.

---

> ### Comment · Reviewer_Ycco · 2022-09-20
> **after rebuttal**
>
> --After authors' response I have read the authors response and the update made on the manuscript. My major concern was on clarifying the methodologies, and I think the authors have made quite significant (and positive) changes as to improving the paper.

---

### Decision · Action_Editors · 2022-09-27

**Recommendation:** Accept with minor revision

**Comment:**

The reviewers' initial comments were mostly cleared up after clarification and rewording.

- The usage of the word "projection" is clearly justified in the single vector case, section 2.2.  In the multi vector case, section 3 gives a well-defined operator, equation (GSP), but it's a bit vague whether it's a projection.  Projection in a linear algebra sense means idempotent (which your operation seems to be) but is defined for a linear operator and a domain that is a sub-space, which does not apply to your operator. In optimization, projection is often used in the sense of "best approximation" (with respect to some norm), as in Eq (3), but that's also unclear in the multi vector case. So maybe state in which precise sense the operation is a "projection", or possibly give it a different kind of name.

- The usage of \citet and \citep is not done correctly. See, e.g., https://bochang.me/blog/posts/latex/

- After equation 1, it's written iff "x(i) = b" but this should be "|x(i)| = b".  That same paragraph uses "i" to both index a component and a set of vectors; please use a different variable for the two meanings.

- "Note that while the l0 norm can also be used directly as a sparsity measure Bolte et al. (2014); Pock & Sabach (2016) although it does not enjoy some of the nice properties of the Hoyer-sparsity measure;"  This is not a complete phrase; remove the "while" or the "although" to make it complete.

- There are still other grammar issues, in addition to the ones I just pointed out. Since my job is not to copy-edit, I just request that the reviewers take another look at the manuscript to fix minor typos and grammar issues before they submit the final version.

**Audience:**

There is a small audience who will be interested in the technical details of the projection itself.  A larger audience will be interested in the idea of Hoyer sparsity (this is not the first paper to use the concept, but it does increase the visibility of the concept), as well as researchers who use non-negative matrix factorization (including researchers in applications such as neuroscience, remote sensing, etc.) and researchers interested in sparsifying neural nets (especially those implementing neural nets on embedded devices).  Overall, this is a nice wide audience.


**Claims Support:**

1. For the first claim, theoretical justification is given for this, and the reviewers seemed universally satisfied. It is also unlikely that the numerical results would be reasonable if there were a gross mistake here.

2. For the second claim, here the writing and numerical experiments are the main justification.  After some clarifications, the reviewers seemed happy with this too, and as associate editor, I also find this to be justified.  The proposed method seems to be useful in non-negative matrix factorization and in neural net sparsification.  For a math paper, the authors have done sufficient numerical justification that their method could have real-world impact.

Hence, overall, I find that this paper is a significant positive contribution to the community and I'm happy to recommend acceptance.

**Main Claims:**

The authors introduce a "projection" problem for finding a set of vectors that have a large average alignment with a set of target vectors while having a small average sparsity in the Hoyer sense. The authors show how to solve this efficiently, and this allows group "projections" onto Hoyer-average-sparsity sets, which has applications to non-negative matrix factorization and neural net sparsification.

1. The first claim is that their mathematical solution to the defined "GSP" problem is correct.
2. The second claim, more implicit, is that this is a useful object to study.

---

> ### Author Response · Authors · 2022-10-15
> **Response to Action Editors**
>
> We thank the action editor for his comments which helped us improve the paper. We are glad the action editor found our paper to be of value to a potentially wide audience. We would also like to thank all the reviewers for their careful feedback during the review process which has helped us to significantly improve our paper during the review process. We finally address the comments made by the action editors below:
>
> 1. "The usage of the word "projection" is clearly justified in the single vector case, section 2.2. In the multi vector case, section 3 gives a well-defined operator, equation (GSP), but it’s a bit vague whether it’s a projection ..."
> > We agree with the editor. For simplicity, we prefer to keep the word ’projection’ (otherwise we would need to change it everywhere in our paper, codes, etc.), and added a remark in section 3.1 to clarify this point:
> >
> > Remark 3.1 (Abuse of terminology). The solution to the problem (GSP) is not a projection, as it does not provide a point within a set closest in some norm to a given point. However, for simplicity, and since it extends the projection in the case of a single vector, see (3) and (4), we abuse the terminology and refer to (GSP) as a projection.
>
> 2. The usage of \ citet and \ citep is not done correctly. See, e.g., https://bochang.me/blog/posts/latex/"
> > We have gone through the paper and fixed these instances in the camera ready version.
> 3. "After equation 1, it’s written iff "x(i) = b" but this should be "|x(i)| = b". That same paragraph uses "i" to both index a component and a set of vectors; please use a different variable for the two meanings."
> > We have corrected this.
> 4. "Note that while the l0 norm can also be used directly as a sparsity measure Bolte et al. (2014); Pock & Sabach (2016) although it does not enjoy some of the nice properties of the Hoyer-sparsity measure;" This is not a complete phrase; remove the "while" or the "although" to make it complete.
> > We have corrected this.
> 5. There are still other grammar issues, in addition to the ones I just pointed out. Since my job is not to copy-edit, I just request that the reviewers take another look at the manuscript to fix minor typos and grammar issues before they submit the final version.
> > We have now read the paper carefully again and fixed some grammar and other typos/errors.